# Orai-mediated calcium entry determines activity of central dopaminergic neurons by regulation of gene expression

**Rishav Mitra**[1], **Shlesha Richhariya**[1,2], **Gaiti Hasan**[1]*

[1]National Centre for Biological Sciences, Tata Institute of Fundamental Research, Bangalore, India; [2]Department of Biology, Brandeis University, Waltham, United States

*For correspondence:
gaiti@ncbs.res.in

Competing interest: The authors declare that no competing interests exist.

**Abstract** Maturation and fine-tuning of neural circuits frequently require neuromodulatory signals that set the excitability threshold, neuronal connectivity, and synaptic strength. Here, we present a mechanistic study of how neuromodulator-stimulated intracellular $Ca^{2+}$ signals, through the store-operated $Ca^{2+}$ channel Orai, regulate intrinsic neuronal properties by control of developmental gene expression in flight-promoting central dopaminergic neurons (fpDANs). The fpDANs receive cholinergic inputs for release of dopamine at a central brain tripartite synapse that sustains flight (Sharma and Hasan, 2020). Cholinergic inputs act on the muscarinic acetylcholine receptor to stimulate intracellular $Ca^{2+}$ release through the endoplasmic reticulum (ER) localised inositol 1,4,5-trisphosphate receptor followed by ER-store depletion and Orai-mediated store-operated $Ca^{2+}$ entry (SOCE). Analysis of gene expression in fpDANs followed by genetic, cellular, and molecular studies identified Orai-mediated $Ca^{2+}$ entry as a key regulator of excitability in fpDANs during circuit maturation. SOCE activates the transcription factor trithorax-like (Trl), which in turn drives expression of a set of genes, including *Set2*, that encodes a histone 3 lysine 36 methyltransferase (H3K36me3). Set2 function establishes a positive feedback loop, essential for receiving neuromodulatory cholinergic inputs and sustaining SOCE. Chromatin-modifying activity of Set2 changes the epigenetic status of fpDANs and drives expression of key ion channel and signalling genes that determine fpDAN activity. Loss of activity reduces the axonal arborisation of fpDANs within the MB lobe and prevents dopamine release required for the maintenance of long flight.

## eLife assessment

In *Drosophila melanogaster*, the SOCE channel Orai is required for the development of flight-promoting dopaminergic neurons. The Hasan laboratory has previously shown that disabling Orai function impairs *Drosophila* flight due to aberrant neuronal development at the pupal stage. In this **fundamental** study, Mitra et al. show that SOCE drives a transcriptional feedback loop via the homeobox transcription factor, 'Trithorax-like' (Trl), and histone modifiers, Set2 and E(z), to regulate the expression of key genes required for the function of dopaminergic flight neurons, including the muscarinic acetylcholine receptor and the inositol 1,4,5-trisphosphate receptor. This **solid** study is carefully performed with validated methodology and most of the analyses are rigorous.

## Introduction

Neural circuitry underlying mature adult behaviours emerges from a combination of developmental gene expression programs and experience-dependent neuronal activity. Holometabolous insects such as *Drosophila* reconfigure their nervous system during metamorphosis to generate circuitry capable of

supporting adult behaviours (*Levine, 1984*). During pupal development, neurons undergo maturation of their electrical properties with a gradual increase in depolarising responses and consequent synaptic transmission (*Hardie et al., 1993*; *Järvilehto and Finell, 1983*). Apart from voltage-gated $Ca^{2+}$ channels and fast-acting neurotransmitters specific to ionotropic receptors, the developing nervous system also uses neuromodulators, which target metabotropic receptors, show slower response kinetics, and are capable of affecting a larger subset of neurons by means of diffusion-aided volumetric transmission (*Taber and Hurley, 2014*). Although neuromodulators alter intrinsic neuronal properties (*Marder, 2012*) by changes in gene expression, the molecular mechanisms through which this is achieved and maintained over developmental timescales needs further understanding.

$Ca^{2+}$ signals generated by neuronal activity can determine neurotransmitter specification (*Spitzer, 2012*), synaptic plasticity (*O'Hare et al., 2022*; *Takechi et al., 1998*), patterns of neurite growth (*Gu and Spitzer, 1995*), and gene expression programs (*Ciceri et al., 2022*; *Rosenberg and Spitzer, 2011*) over developmental timescales (*Arjun McKinney et al., 2022*) where output specificity is defined by signal dynamics. Neuromodulators generate intracellular $Ca^{2+}$ signals by stimulation of cognate G-protein-coupled receptors (GPCRs) linked to the inositol 1,4,5-trisphosphate ($IP_3$) and $Ca^{2+}$ signalling pathway. $IP_3$ transduces intracellular $Ca^{2+}$ release through the endoplasmic reticulum (ER) localised ligand-gated ion channel, the inositol 1,4,5-trisphosphate receptor ($IP_3R$; *Streb et al., 1984*), followed by store-operated $Ca^{2+}$ entry (SOCE) through the plasma membrane localised Orai channel (*Prakriya and Lewis, 2015*; *Thillaiappan et al., 2019*). Cellular and physiological consequences of $Ca^{2+}$ signals generated through SOCE exhibit different timescales and subcellular localisation from activity-induced signals, suggesting that they alter neuronal properties through novel mechanisms.

Expression studies, genetics, and physiological analysis support a role for $IP_3/Ca^{2+}$ and SOCE in neuronal development and the regulation of adult neuronal physiology across organisms (*Hasan and Sharma, 2020*; *Mitra and Hasan, 2022*; *Somasundaram et al., 2014*). In *Drosophila*, neuromodulatory signals of neurotransmitters and neuropeptides stimulate cognate GPCRs on specific neurons to generate $IP_3$ followed by intracellular $Ca^{2+}$ release and SOCE (*Agrawal et al., 2013*; *Megha and Hasan, 2017*; *Shakiryanova et al., 2011*). Genetic and cellular studies identified neuromodulatory inputs and SOCE as an essential component for maturation of neural circuits for flight (*Pathak et al., 2015*; *Venkiteswaran and Hasan, 2009*). A focus of this flight deficit lies in a group of central dopaminergic neurons (DANs) that include the PPL1 and PPM3 DANs (*Liu et al., 2012*) marked by *THD' GAL4*, alternately referred to henceforth as flight-promoting DANs (fpDANs). Among the fpDANs, some PPL1 DANs project to the γ2α'1 lobe of the mushroom body (MB; *Mao and Davis, 2009*), a dense region of neuropil in the insect central brain, that forms a relay centre for flight (*Sharma and Hasan, 2020*). Axonal projections from Kenyon cells (KCs) carry sensory inputs (*Cervantes-Sandoval et al., 2017*; *Tsao et al., 2018*; *Yagi et al., 2016*) and those from DANs carry internal state information (*Mao and Davis, 2009*; *Riemensperger et al., 2005*; *Zolin et al., 2021*) to functional compartments of the MB, where dopamine release modulates the KC and MB output neuron synapse (MBONs; *Aso et al., 2014*). The KC-DAN-MBON tripartite synapse carries a dynamically updating representation of the motivational and behavioural state of the animal (*Aso and Rubin, 2016*; *Berry et al., 2015*; *Owald et al., 2015*; *Waddell, 2016*). Here, we have investigated the molecular mechanisms that underlie the ability of neuromodulatory acetylcholine signals, acting through SOCE during circuit maturation, to determine fpDAN function required for *Drosophila* flight.

## Results

### A spatio-temporal requirement for SOCE determines flight

To understand how neuromodulator-stimulated SOCE alters neuronal properties over developmental timescales, we began by refining further the existing spatio-temporal coordinates of SOCE requirement for flight. fpDANs with a requirement for SOCE have been identified earlier by expression of a dominant negative *Orai* transgene, which renders the channel $Ca^{2+}$ impermeable, thus abrogating all SOCE as observed in primary neuronal cultures (*Orai*$^{E180A}$; *Figure 1A*; *Pathak et al., 2015*; *Yeromin et al., 2006*). These include a smaller subset of *THD'GAL4* marked DANs, where acetylcholine promotes dopamine release through the muscarinic acetylcholine receptor (mAchR) by stimulating $IP_3/Ca^{2+}$ signalling (*Sharma and Hasan, 2020*). *Orai*$^{E180A}$ expression in a subset of 21–23 DANs

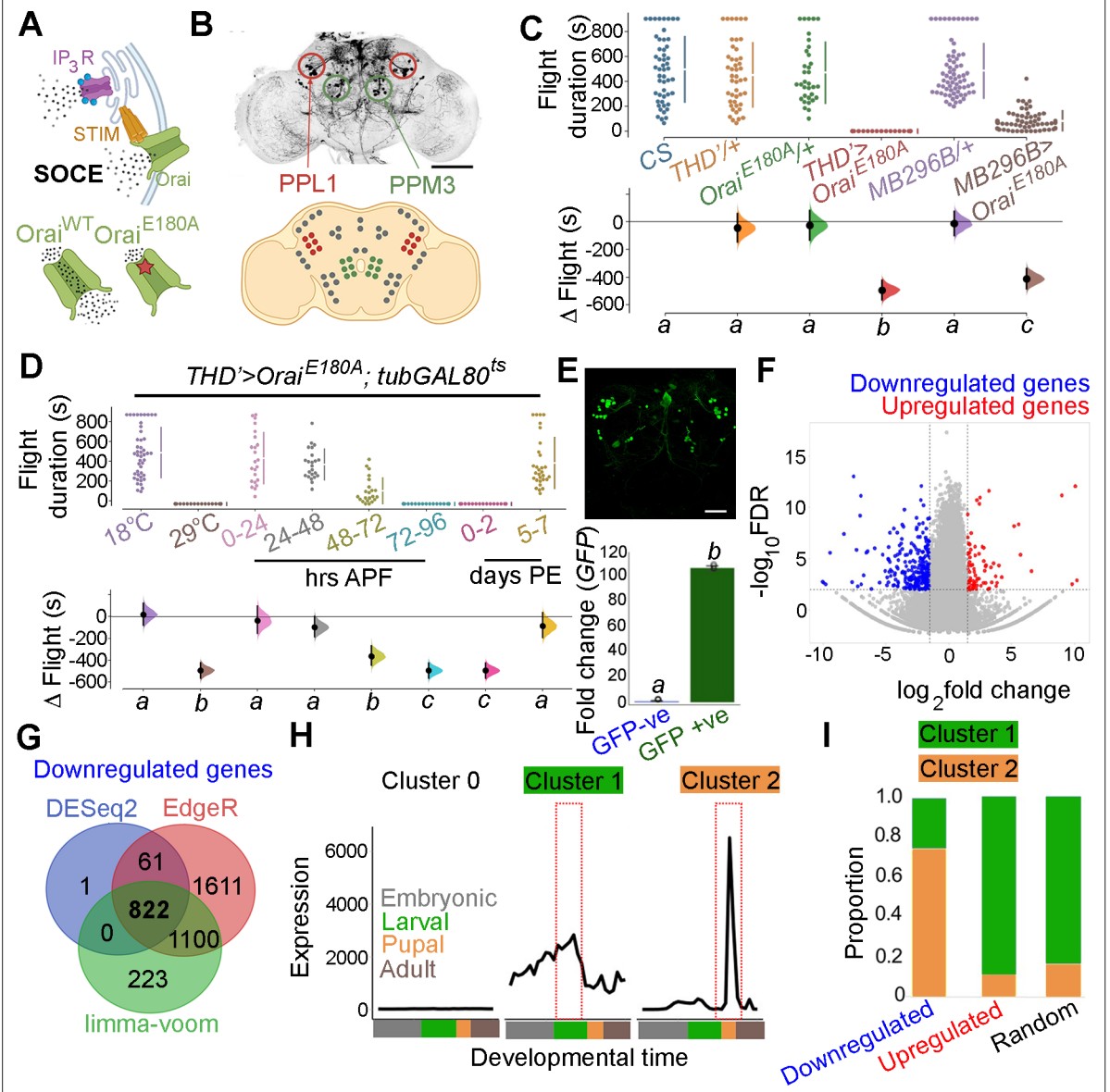

**Figure 1.** Orai-mediated Ca²⁺ entry sets the gene expression profile of flight-promoting dopaminergic neurons (DANs) in late development and early adulthood. (**A**) A schematic of Ca²⁺ release through the inositol 1,4,5-trisphosphate receptor (IP₃R) and store-operated Ca²⁺ entry (SOCE) through STIM/Orai (upper panel), followed by representation of the wildtype (Ca²⁺ permeable) and mutant (Ca²⁺ impermeable) Orai channels (lower panel). (**B**) Anatomical location of *THD'* DANs in the fly central brain immunolabelled for mCD8GFP (upper panel), followed by a cartoon of central brain DAN clusters. Scale bar indicates 20 µm. PPL1 and PPM3 clusters are labelled in red and green, respectively. (**C**) Measurement of flight bout durations demonstrates a requirement for Orai-mediated Ca²⁺ entry in *THD'* DANs and in two pairs of PPL1 DANs marked by the MB296B driver. (**D**) *THD'* DANs require Ca²⁺ entry through Orai at 72–96 hr after puparium formation (APF) and 0–2 d post eclosion to promote flight. In (**C**) and (**D**), flight bout durations in seconds (s) are represented as a swarm plot where each genotype is represented by a different colour, and each fly as a single data point. The Δ Flight parameter shown below indicates the mean difference for comparisons against the shared Canton S control and is shown as a Cumming estimation plot. Mean differences are plotted as bootstrap sampling distributions. Each 95% confidence interval is indicated by the ends of the vertical error bars. The same letter beneath each distribution refers to statistically indistinguishable groups after performing a Kruskal–Wallis test followed by a post hoc Mann–Whitney U-test (p<0.005). At least 30 flies were tested for each genotype. (**E**) *THD'* DANs were labelled with cytosolic eGFP (10 µm scale), isolated using fluorescence-activated and sorted (FACS), and validated for enrichment of *GFP* mRNA by qRT-pCR (lower panel). The qRT-pCR results are from four biological replicates, with different letters representing statistically distinguishable groups after performing a two-tailed *t*-test (p<0.05). (**F**) RNA-seq comparison of FAC-sorted populations of GFP-labelled THD' DANs from *THD'>GFP* and *THD'>GFP;Orai^{E180A}* pupal dissected central nervous systems (CNSs). The RNA-seq data is represented in the form of a volcano plot of fold change vs FDR. Individual dots represent genes, coloured in red (upregulated) or blue (downregulated) by greater than onefold. (**G**) Downregulated genes were identified by three different methods of differentially expressed gene (DEG) analysis, quantified, and compared as a Venn diagram. (**H**) Gene expression trajectories of SOCE-induced DEGs

*Figure 1 continued on next page*

*Figure 1 continued*

plotted as a function of developmental time (*modENCODE Consortium et al., 2010*) and clustered into three groups using k-means analysis. (**I**) The relative proportion of downregulated, upregulated genes, and a random set of genes found in the clusters described in (**H**) indicates that 75% of downregulated genes exhibit a pupal peak of expression.

The online version of this article includes the following source data and figure supplement(s) for figure 1:

**Source data 1.** Raw data for flight assays (C, D), IHC quantification (E), and differential gene expression analysis (F, H, I).

**Figure supplement 1.** Flight assays demonstrate that overexpression of *Orai* in *THD'* dopaminergic neurons (DANs) does not cause severe flight deficits.

**Figure supplement 1—source data 1.** Raw data for flight assays.

**Figure supplement 2.** A schematic representation (upper left) of the different lobes of the mushroom body, and a table denoting split GAL4 lines marking different dopaminergic neuron (DAN) subsets sending projections to them (bottom).

**Figure supplement 3.** Testing the requirement of Orai function in different dopaminergic neuron (DAN) subsets described in *Figure 1—figure supplement 2*.

**Figure supplement 3—source data 1.** Raw data for flight assays.

**Figure supplement 4.** Selective inhibition of *THD'* neuronal activity using *Shibire*[ts] or pupal-specific Tetanus toxin (*TeTxLC; GAL80*[ts]) causes a significant loss in flight bout durations, indicating that these neurons are required for flight.

**Figure supplement 4—source data 1.** Raw data for flight assays.

**Figure supplement 5.** Fluorescence-activated and sorted (FACS) plot for *THD'>eGFP*, where each dot represents isolated neurons from the brain, with relevant gates for purification of GFP-expressing cells.

**Figure supplement 6.** Venn diagram representing the intersections of genes upregulated upon loss of *Orai* function identified using three different methods of analysis.

**Figure supplement 7.** Mean gene expression trajectories of upregulated, downregulated, or a random set of genes over developmental timescales (data from *Kim et al., 2019*).

**Figure supplement 8.** Loss of *Orai* function results in genes being flipped in their expression state.

**Figure supplement 8—source data 1.** Raw data for transcript quantification from RNA-Seq.

**Figure supplement 9.** A kernel density estimation plot of baseline expression of up- or downregulated genes indicates that downregulated genes have higher baseline expression levels.

**Figure supplement 9—source data 1.** Raw data for transcript quantification from RNA-Seq.

marked by *THD'* GAL4 (*Figure 1B*) led to complete loss of flight (*Figure 1C*) while overexpression of the wildtype *Orai* transgene had a relatively minor effect (*Figure 1—figure supplement 1*).

*THD'* marks DANs in the PPL1, PPL2, and PPM3 clusters of the adult central brain (*Figure 1B*) of which the PPL1 (10–12) and PPM3 (6–7) cell clusters constitute the fpDANs (*Pathak et al., 2015*). Among these 16–19 DANs, two pairs of PPL1 DANs projecting to the γ2α'1 lobes of the MB (marked by the MB296B split GAL4 driver and schematised in *Figure 1—figure supplement 2*; *Aso et al., 2014*; *Aso and Rubin, 2016*) were identified as contributing to the flight deficit to a significant extent (*Figure 1C*, *Figure 1—figure supplement 3*). The MB296B DANs form part of a central flight relay centre identified earlier as requiring IP$_3$/Ca$^{2+}$ signalling (*Sharma and Hasan, 2020*). That the *THD'* neurons are required for flight was further confirmed by inhibiting their function. Acute activation of either a temperature-sensitive *Dynamin* mutant (*Shibire*[ts]; *Kitamoto, 2001*), which blocks exocytosis at an elevated temperature, or acute expression of the tetanus toxin light chain fragment (*TeTxLC*; *Sweeney et al., 1995*), which cleaves *Synaptobrevin*, an essential component of the synaptic release machinery in *THD'* neurons, led to severe flight defects (*Figure 1—figure supplement 4*).

Previous work has demonstrated that intracellular Ca$^{2+}$ release through the IP$_3$R, which precedes SOCE (*Figure 1A*), is required during late pupal development in *THD'*-marked DANs for flight (*Pathak et al., 2015*; *Sharma and Hasan, 2020*). The precise temporal requirement for SOCE in *THD'* neurons was investigated by inducing *Orai*[E180A] transgene expression for specific periods of pupal development and in adults with the *tubulinGAL80*[ts]-based temperature-sensitive expression system (TARGET – Temporal And Regional Gene Expression Targeting; *McGuire et al., 2004*). Approximately 100 hr of pupal development were binned into 24 hr windows, wherein SOCE was abrogated by *Orai*[E180A] expression, following which normal development and growth were permitted. Flies were assayed for flight 5 d post eclosion. Abrogation of SOCE by expression of the *Orai*[E180A]-dominant-negative transgene at 72–96 hr after puparium formation (APF) resulted in complete loss of flight compared

to minor flight deficits observed upon expression of $Orai^{E180A}$ during earlier developmental windows (*Figure 1D*). Abrogation of flight also occurred upon $Orai^{E180A}$ expression during the first 2 d post-eclosion (see 'Discussion'), whereas abrogation of SOCE 5 d after eclosion as adults resulted in only modest flight deficits (*Figure 1D*), indicating that SOCE is required for maturation of *THD'*-marked flight DANs. Earlier work has shown that loss of SOCE by expression of $Orai^{E180A}$ does not alter the number of DANs or affect neurite projection patterns of PPL1 and PPM3 neurons (*Pathak et al., 2015*). Of these two clusters, PPM3 DANs project to the ellipsoid body in the central complex (*Kong et al., 2010*), and PPL1 DANs project to the MB and the lateral horn (*Mao and Davis, 2009*) and aid in the maintenance of extended flight bouts (*Sharma and Hasan, 2020*).

## Loss of SOCE in late pupae leads to a reorganisation of neuronal gene expression

The molecular consequences of loss of SOCE in *THD'* neurons were investigated next by undertaking a comprehensive transcriptomic analysis from fluorescence-activated and sorted (FACS) *THD'* neurons with or without $Orai^{E180A}$ expression and marked with eGFP (*Figure 1E*, *Figure 1—figure supplement 5*). *THD'* neurons were obtained from pupae at 72 ± 6 hr APF (essential SOCE requirement for flight; *Figure 1E*). Expression analysis of RNA-seq libraries generated from *THD'* neurons revealed a reorganisation of the transcriptome upon loss of SOCE (*Figure 1F*). Expression of 822 genes was downregulated (with a fold change cut-off of < –1) as assessed using three different methods of differentially expressed genes (DEG) analysis (*Figure 1G*), whereas 137 genes were upregulated (*Figure 1—figure supplement 6*). To understand whether loss of SOCE affects genes expressed in neurons from the pupal stage, we used the modEncode dataset (*Celniker et al., 2009*; *modENCODE Consortium et al., 2010*) to reconstruct developmental trajectories of all genes expressed in *THD'* neurons (*Figure 1—figure supplement 7*). Using an unsupervised clustering algorithm, we classified genes expressed in *THD'* neurons into three clusters, where 'cluster 0' is low expression throughout development, 'cluster 1' exhibits a larval peak in expression, and 'cluster 2' exhibits a pupal peak in expression (*Figure 1H*). The majority of expressed genes (>95%) classify as low expression throughout (cluster 0) and were removed from further analysis. Genes belonging to either the downregulated gene set or the upregulated gene set in SOCE-deficient *THD'* neurons and a random set of genes were analysed further. A comparison of the proportion of DEGs classified in the larval and pupal clusters revealed that >75% of downregulated genes belonged to the pupal peak (cluster 2), whereas <20% of either the upregulated genes or a random set of genes classified as belonging to the larval peak (*Figure 1I*). Moreover, downregulated genes exhibit higher baseline expression in wildtype pupal *THD'* neurons (*Figure 1—figure supplements 8 and 9*). These analyses suggest that SOCE induces expression of a set of genes in *THD'* neurons during the late pupal phase, which are subsequently required for flight. These are henceforth referred to as SOCE-responsive genes.

## SOCE regulates gene expression through a balance of histone 3 lysine 36 trimethylation and histone 3 lysine 27 trimethylation

Gene Ontology (GO) analysis of SOCE-responsive genes revealed 'Transcription', 'Ion transporters', '$Ca^{2+}$ dependent exocytosis', 'GPCRs', 'Synaptic components', and 'Kinases' as top GO categories (*Figure 2A*). Among these, 'Transcription' and 'Ion transporters' represented categories with the highest enrichment. Further analysis of genes that classified under 'Transcription' revealed SET domain-containing genes that encode histone lysine methyltransferases implicated in chromatin regulation and gene expression (*Dillon et al., 2005*), and among the SET domain-containing genes, *Set1* and *Set2* showed distinct downregulation (*Figure 2B*). One of these genes, the H3K36me3 methyltransferase *Set2* (*Figure 2C*), was identified in a previous RNA-seq from larval neurons with loss of $IP_3R$-mediated $Ca^{2+}$ signalling (*Mitra et al., 2020*). RT-PCRs from sorted *THD'* neurons confirmed that *Set2* levels are indeed downregulated (by 76%) in *THD'* neurons expressing $Orai^{E180A}$ and to the same extent as seen upon expression of $Set2^{RNAi}$ (79% downregulation; *Figure 2C*). *Set1* and *Set2* perform methyltransferase activity at H3K4 and H3K36 histone residues, respectively (*Ardehali et al., 2011*; *Stabell et al., 2007*). Both these markers are epigenetic signatures for transcriptional activation, where H3K4me3 is enriched at the 5' end of the gene body and H3K36me3 is enriched in the gene bodies of actively transcribed genes; schematised in *Figure 2—figure supplement 1*. In the adult central nervous system (CNS), *Set2*, is expressed at fourfold higher levels compared to

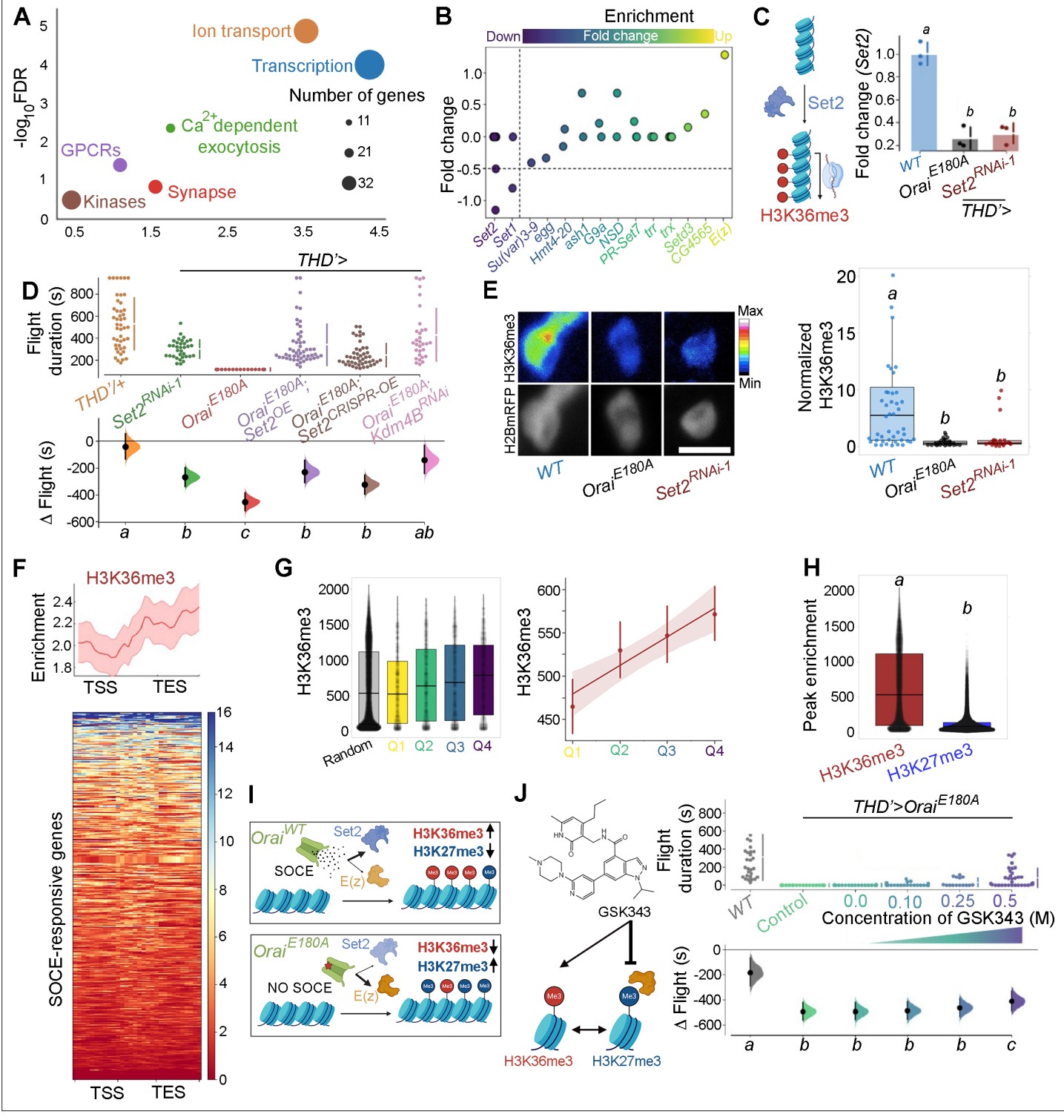

**Figure 2.** Ca$^{2+}$ entry through Orai regulates gene expression by Set2-mediated histone modification. (**A**) Scatter plot of Gene Ontology (GO) categories enriched in store-operated Ca$^{2+}$ entry (SOCE)-responsive genes. Individual GO terms are represented as differently coloured circles, with radius size indicating number of genes enriched in that category. (**B**) Fold change of SET domain containing genes as indicated. Individual circles on the Y-axis for each gene represent transcript variants pertaining to that gene. (**C**) Transcripts of the H3K36 methyltransferase (left) are significantly diminished (right) in *THD'* dopaminergic neurons (DANs) either upon loss of Orai function (*THD'>OraiE$^{E180A}$*) or by knockdown of *Set2* (*THD'>Set2* RNAi). qRT-PCRs were performed from FAC-sorted *THD'* DANs with three biological replicates. Individual 2$^{-\Delta\Delta CT}$ values are shown as points. Letters represent statistically distinguishable groups after performing an ANOVA and post hoc Tukey test (p<0.05). (**D**) Significant rescue of flight bout durations seen

*Figure 2 continued on next page*

*Figure 2 continued*

in *THD'>OraiE^{E180A}* flies by overexpression of *Set2* and by knockdown of the *Kdm4B* demethylase indicating a net requirement for H3K36me3. Flight durations of single flies are depicted as swarm plots and the Δ flight parameter is shown below. Both were measured as described in the legend to *Figure 1*. N = 30 or more flies for each genotype. Letters represent statistically distinguishable groups after performing an ANOVA and post hoc Tukey test (p<0.005). (E) Representative images (upper panel) and quantification (lower panel) from immunostaining of H3K36me3 and H2BmRFP in nuclei of *THD'* DANs from at least 10 brains. Scale bar represents 5 μm. The boxplot represents individual H3K36me3/H2BmRFP ratios from each *THD'* DAN for each genotype. Letters represent statistically distinguishable groups after performing an ANOVA and post hoc Tukey test (p<0.05). (F) H3K36me3 enrichment over the gene bodies of SOCE-responsive genes in wildtype fly heads represented in the form of a tag density plot. (G) H3K36me3 signal is enriched on WT SOCE-responsive genes with greater downregulation upon loss of Orai function. Individual data points represent *WT* H3K36me3 ChIP-seq signals from adult fly heads represented as a boxplot (left) and a regression plot (right; Pearson's correlation coefficient = 0.11), indicating a correlation between extent of downregulation upon loss of SOCE and greater enrichment of H3K36me3. (H) SOCE-responsive genes are enriched in H3K36me3 signal compared to H3K27me3 signal as measured from relevant ChIP-seq datasets. Adult fly head ChIP-seq datasets for measurements in (F–H) were obtained from modEncode (*modENCODE Consortium et al., 2010*). (I) Schematic representation of how Orai-mediated Ca^{2+} entry regulates a balance of two opposing epigenetic signatures in developing DANs. (J) Pharmacological inhibition of H3K27me3 using GSKS343 (left) in *THD'>OraiE^{E180A}* flies results in a dose-dependent rescue of flight bout durations (right). Flight assay measurements from N > 30 flies, are represented as described earlier. Letters represent statistically distinguishable groups after performing an ANOVA and post hoc Tukey test (p<0.005).

The online version of this article includes the following source data and figure supplement(s) for figure 2:

**Source data 1.** Raw data for GO Analysis (A), Transcript quantification (B), RT-qPCRs (C), flight assays (D, J), IHC quantification (E), and ChIP-Seq peak quantification (F, G, H).

**Figure supplement 1.** Schematic comparison between Set2-mediated H3K36me3 (red) and Set1-mediated H3K4me3 (cyan) over the gene body.

**Figure supplement 2.** Developmental gene expression trajectories of *Set1* and *Set2* (modEncode).

**Figure supplement 2—source data 1.** Raw data for transcript quantification from RNA-Seq.

**Figure supplement 3.** *Set2* is required in *THD'* dopaminergic neurons (DANs) for flight.

**Figure supplement 3—source data 1.** Raw data for flight assays.

**Figure supplement 4.** Set2 is required in mushroom body dopaminergic neurons (DANs) relevant for flight.

**Figure supplement 4—source data 1.** Raw data for flight assays.

**Figure supplement 5.** Flight defects in the *Orai* loss-of-function background could not be rescued by overexpression of a control *GCaMP6m* transgene.

**Figure supplement 5—source data 1.** Raw data for flight assays.

**Figure supplement 6.** Of the two H3K36 demethylases, *Kdm4B* is more highly expressed in *THD'* dopaminergic neurons (DANs) at 72 hr after puparium formation (APF).

**Figure supplement 6—source data 1.** Raw data for transcript quantification from RNA-Seq.

**Figure supplement 7.** *Kdm4B* but not *Kdm4A* overexpression in *THD'* dopaminergic neurons (DANs) mediates a flight rescue in the *Orai* loss-of-function background.

**Figure supplement 7—source data 1.** Raw data for flight assays.

**Figure supplement 8.** GSK343 feeding results in a dose-dependent rescue in terms of number of flies showing >30 s of flight.

**Figure supplement 8—source data 1.** Raw data for flight assays.

*Set1* (*Figure 2—figure supplement 2*; *modENCODE Consortium et al., 2010*). *Set2* also shows a steep 5.5-fold increase in expression levels during the pupal to adult transition compared to a 2-fold increase in *Set1* (*Figure 2—figure supplement 2*). These observations, coupled with the previous report of Set2 function downstream of IP₃R/Ca^{2+} signalling in larval glutamatergic neurons (*Mitra et al., 2020*), led us to pursue the role of Set2 in *THD'* neurons.

Set2 encodes the only *Drosophila* methyltransferase which is specific for H3K36 trimethylation (*Stabell et al., 2007*). To understand the functional significance, if any, of downregulation of *Set2* in *THD'* neurons upon loss of SOCE, four independent *Set2* RNAi constructs were obtained. Knockdown of *Set2* in *THD'* neurons with all four RNAi lines tested resulted in significant flight defects (*Figure 2D*, *Figure 2—figure supplement 3*). *Set2* function for flight was additionally verified in the PPL1 DANs projecting to the γ2α'1 lobe of the MB (*MB296BGAL4*; *Figure 2—figure supplement 4*). To test the hypothesis that *Set2* expression and function for flight is regulated by SOCE, we overexpressed *Set2* in the *Orai^{E180A}* background using GAL4-UAS-driven heterologous *Set2^{WT}* expression or CRISPR-dCas9::VPR-driven overexpression (*Gilbert et al., 2013*) from the endogenous locus and measured flight (*Figure 2D*, data in purple and brown). Flies expressing *Orai^{E180A}* in the

*THD'* neurons, which are flightless, exhibit significant rescue in flight bout durations upon *Set2* over-expression, through either method (**Figure 2D**, compare red with purple and brown data), but not upon expression of a control transgene, *GCamP6m* (**Figure 2—figure supplement 5**, compare data in green and red). Although the *Set2* overexpression rescues were induced all through development, because Orai function drives critical gene expression during 72–96 hr APF (**Figure 1D**), we concluded that Set2-mediated rescue is also likely to occur during this time window. These data taken together with downregulation of *Set2* upon expression of *Orai$^{E180A}$* (**Figure 2C**) support the hypothesis that SOCE-driven expression of *Set2* in fpDANs is required for flight. That Set2 function downstream of SOCE occurs specifically through its methyltransferase activity was supported through experiments where knockdown of the demethylase *Kdm4B*, the primary histone demethylase expressed in these neurons (**Figure 2—figure supplement 6**), rescued flight in *THD'>OraiE$^{E180A}$* flies (**Figure 2D**, data in pink). Knockdown of the lower expressed *Drosophila* H3K36 demethylase isoform, *Kdm4A,* did not show a similar rescue (**Figure 2—figure supplement 7**, data in pink). Moreover, both *Set2* depletion and loss of SOCE (*THD'>OraiE$^{E180A}$*) lead to a decrease in the normalised H3K36me3 signal in *THD'* neurons (**Figure 2E**).

To understand whether deficient H3K36me3 is found in the SOCE-responsive genes, we quantified the enrichment of H3K36me3 at SOCE-responsive loci from an H3K36me3 ChIP-seq dataset (**modEN-CODE Consortium et al., 2010**) from wildtype adult heads (**Figure 2F**). We observed an enrichment of the H3K36me3 signal over the gene bodies of the SOCE-responsive genes (**Figure 2F**). Addition-ally, genes that were more affected by loss of SOCE (greater extent of downregulation) had a corre-spondingly higher H3K36me3 signal (**Figure 2G**) compared with a random set of genes (**Figure 2G**, data in gray), further indicating that one of the ways SOCE regulates gene expression is through Set2-mediated deposition of H3K36me3.

The gene *E(z)* that encodes a component of the PRC2 complex (*EZH2*) which deposits H3K27me3 is upregulated upon loss of SOCE (**Figure 2B**), suggesting that SOCE could affect additional histone modifications. While H3K36me3 is a marker for transcriptional activation (**Krogan et al., 2003**), H3K27me3 is a repressive signature which silences transcription (**Cai et al., 2021**). These two marks are antagonistic as deduced from studies which have shown that deposition of H3K36me3 allosterically inhibits the Polycomb Repressive Complex (PRC2), thereby preventing it from depos-iting the H3K27me3 signature at the same genomic loci (**Finogenova et al., 2020**). A comparison of H3K36me3 and H3K27me3 peaks over the 822 SOCE-responsive genes revealed a higher enrichment of H3K36me3 versus H3K27me3 in wildtype fly heads (**Figure 2H**; **modENCODE Consortium et al., 2010**), suggesting that robust expression of these genes in adult neurons occurs through an SOCE-dependent mechanism initiated in late pupae (schematised in **Figure 2I**). To test this hypothesis, we fed SOCE-deficient flies (*THD'GAL4>OraiE$^{E180A}$*) GSK343, a pharmacological inhibitor (**Verma et al., 2012**) of the EZH2 component of PRC2 that is known to reduce H3K27me3 chromatin marks and performed flight assays. Feeding of GSK343 lead to partial rescue of flight in *THD'>OraiE$^{E180A}$* flies in a dose-dependent manner with maximal flight bout durations of up to 400 s (**Figure 2J**, **Figure 2—figure supplement 8**).

## Normal cellular responses of PPL1 DANs require Orai-mediated SOCE and H3K36 trimethylation

Next we investigated Ca$^{2+}$ responses in SOCE and H3K36 trimethylation-deficient *THD'* PPL1 DANs upon stimulation with a neuromodulatory signal for the mAChR. It is known that *THD'*-marked PPL1 DANs receive cholinergic inputs that stimulate store Ca$^{2+}$ release through the IP$_3$R, leading to SOCE through Orai (**Figure 3A**; **Ebihara et al., 2006**; **Sharma and Hasan, 2020**). In support of this idea, either expression of *Orai$^{E180A}$* or knockdown of the IP$_3$R attenuated the composite ER-Ca$^{2+}$ release plus SOCE response to carbachol (CCh), a mAChR agonist, in PPL1 DANs (**Figure 3B–D**, **Figure 3—figure supplement 1**). Ex vivo brain preparations used here for in situ Ca$^{2+}$ imaging of fpDANs are not viable in the absence of extracellular Ca$^{2+}$. Hence, measurement of store-Ca$^{2+}$ release and SOCE independent of each other was not possible. Importantly, knockdown of *Set2* also abrogated the Ca$^{2+}$ response to CCh (**Figure 3B–D**), whereas overexpression of a transgene encoding *Set2* in *THD'* neurons either with loss of SOCE (*Orai$^{E180A}$*) or with knockdown of the IP$_3$R (*itpr$^{RNAi}$*) leads to significant rescue of the Ca$^{2+}$ response, indicating that *Set2* is required downstream of SOCE (**Figure 3B–D**) and IP$_3$/Ca$^{2+}$ signalling (**Figure 3—figure supplement 1**).

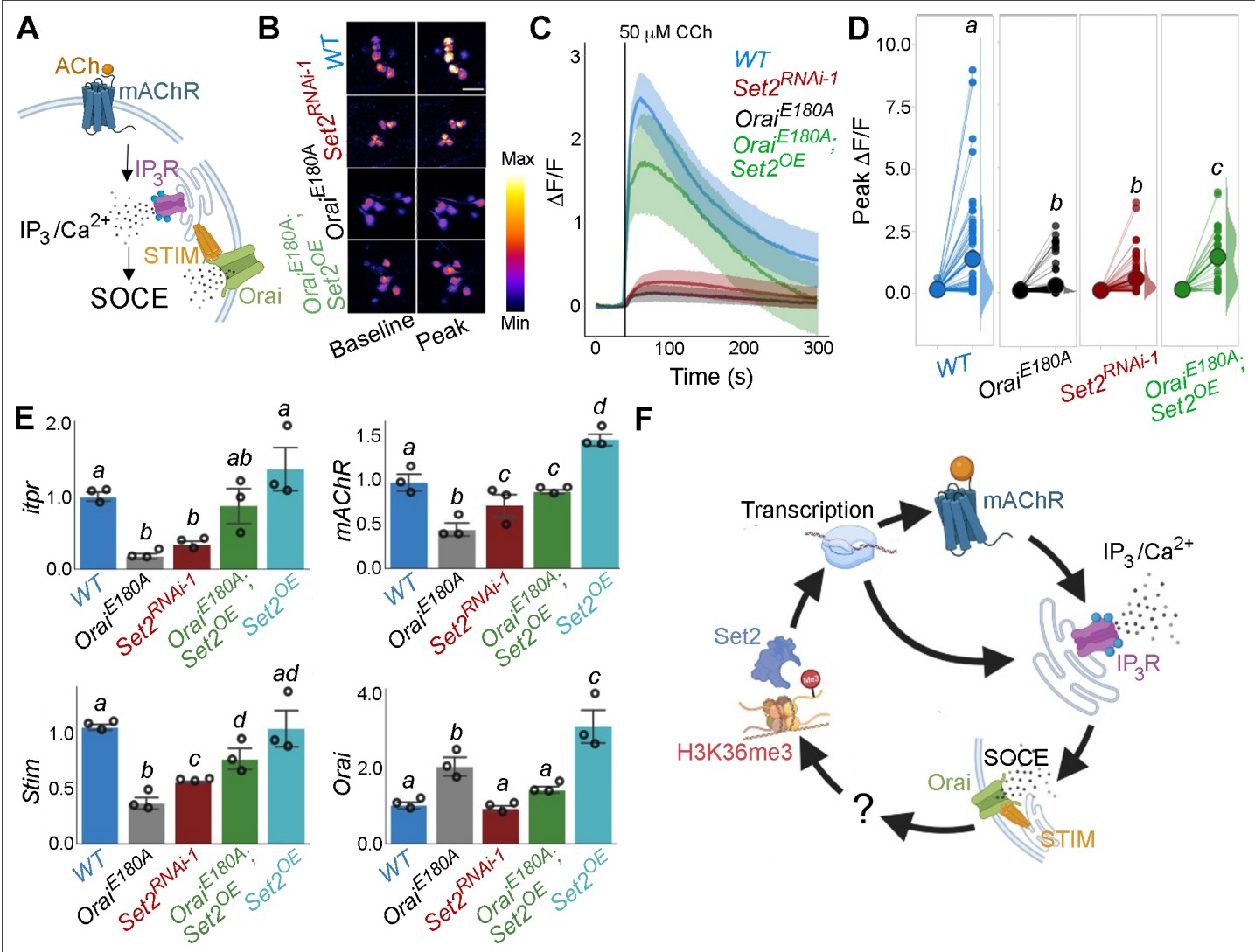

**Figure 3.** Orai-mediated $Ca^{2+}$ entry potentiates cellular $Ca^{2+}$ responses to cholinergic inputs through Set2 and a transcriptional feedback loop. (**A**) A schematic of intracellular $Ca^{2+}$ signalling downstream of neuromodulatory signalling where activation of muscarinic acetylcholine receptor (mAchR) stimulates intracellular $Ca^{2+}$ release through the inositol 1,4,5-trisphosphate receptor ($IP_3R$) followed by store-operated $Ca^{2+}$ entry (SOCE) through STIM/Orai. (**B**) Cholinergic inputs by addition of carbachol (CCh) evoke $Ca^{2+}$ signals as measured by change in the fluorescence of GCaMP6m in *THD'* dopaminergic neurons (DANs) of ex vivo brains. Representative GCaMP6m images of *THD'* DANs are shown with baseline and peak-evoked responses in the indicated genotypes. Scale bar = 10 µM. (**C**) Median GCaMP6m responses plotted as a function of time. A shaded region around the solid line represents the 95% confidence interval from 4 to 5 cells imaged per brain from 10 or more brains per genotype. (**D**) Individual cellular responses depicted as a paired plot where different letters above indicate statistically distinguishable groups after performing a Kruskal–Wallis test and a Mann–Whitney *U*-test (p<0.05). (**E**) qRT-PCR measurements of *itpr*, *mAChR*, *Stim*, and *Orai* from FAC-sorted *THD'* DANs obtained from three biological replicates of appropriate genetic backgrounds. The genotypes include DANs with loss of cellular $Ca^{2+}$ responses (*THD'>OraiE180A*; and *THD'>Set2RNAi-1*), rescue of cellular $Ca^{2+}$ response by overexpression of *Set2* (*THD'>OraiE180A*; *Set2OE*), and *Set2* overexpression. Bar plots indicate mean expression levels in comparison to *rp49*, with individual data points represented as hollow circles. The letters above indicate statistically indistinguishable groups after performing a Kruskal–Wallis test and a Mann–Whitney *U*-test (p<0.05). (**F**) Schematic representation of a transcriptional feedback loop downstream of cholinergic stimulation in *THD'* DANs.

The online version of this article includes the following source data and figure supplement(s) for figure 3:

**Source data 1.** Raw data for imaging quantification (C, D) and RT-qPCRs (E).

**Figure supplement 1.** Carbachol (CCh)-evoked $Ca^{2+}$ responses require inositol 1,4,5-trisphosphate receptor ($IP_3R$) function.

**Figure supplement 1—source data 1.** Raw data for imaging quantification (B, C).

An understanding of how Set2 might rescue cellular IP$_3$/Ca$^{2+}$ signalling was obtained in an earlier study where it was demonstrated that *Set2* participates in a transcriptional feedback loop to control the expression of key upstream components such as the *mAChR* and the *IP$_3$R* in a set of larval glutamatergic neurons (*Mitra et al., 2020*). To test whether *Set2* acts through a similar transcriptional feedback loop in the current context, *THD'* DANs were isolated using FACS from appropriate genetic backgrounds and tested for expression of key genes required for IP$_3$/Ca$^{2+}$ signalling (*mAChR* and *itpr*) and SOCE (*Stim* and *Orai*). Loss of SOCE upon expression of *Orai$^{E180A}$* leads to a significant decrease in the expression of *itpr* (72% downregulation; *Figure 3E*) and *mAChR* (56% downregulation; *Figure 3E*) in *THD'* DANs. As expected, *Orai* expression (*Figure 3E*) increased twofold presumably due to overexpression of the *Orai$^{E180A}$* transgene. Importantly, knockdown of *Set2* in *THD'* neurons also led to downregulation of *itpr* (43%), *mAChR* (78%), and *Stim* (61%) (*Figure 3E*). Overexpression of *Set2* in the background of *Orai$^{E180A}$* led to a rescue in the levels of *itpr* (69% increase), *mAChR* (43% increase), and *Stim* (40% increase) (*Figure 3E*). Overexpression of *Set2* in wildtype *THD'* neurons resulted in upregulation of *mAChR* and *Orai* (*Figure 3E*). These findings indicate that while SOCE regulates *Set2* expression (*Figure 2C*), *Set2* in turn functions in a feedback loop to regulate expression of key components of IP$_3$/Ca$^{2+}$ (*mAChR* and *IP$_3$R*) and SOCE (*STIM* and *Orai*) (*Figure 3E*). Thus, ectopic *Set2* overexpression in SOCE-deficient *THD'* neurons leads to increase in expression of key genes that facilitate intracellular Ca$^{2+}$ signalling and SOCE (schematised in *Figure 3F*). Direct measures of Orai-channel function under conditions of altered *Set2* expression are needed in future to assess how the feedback loop alters CCh-induced ER-Ca$^{2+}$ release and SOCE independent of each other (see 'Limitations of this study'). While we assume that the observed changes in gene expression translate to alterations in protein levels, direct measurement of protein levels specifically from *THD'* neurons needs to be addressed in the future.

## Trithorax-like (Trl) is an SOCE-responsive transcription factor in *THD'* DANs

The data so far identify SOCE as a key developmental regulator of the neuronal transcriptome in *THD'*-marked DANs, where it upregulates *Set2* expression and thus enhances the activation mark of H3K36 trimethylation on specific chromatin regions. However, the mechanism by which SOCE regulates expression of *Set2* and other relevant effector genes remains unresolved (*Figure 3F*). In mammalian T cells, SOCE leads to de-phosphorylation of the NFAT (nuclear factor of activated T cells) family of transcription factors (TFs) by Ca$^{2+}$/calmodulin-sensitive phosphatase calcineurin, followed by their nuclear translocation and ultimately transcription of relevant target genes (*Hogan et al., 2003*). Unlike mammals, the *Drosophila* genome encodes a single member of the NFAT gene family, which does not possess calcineurin binding sites, and is therefore insensitive to intracellular Ca$^{2+}$ and SOCE (*Keyser et al., 2007*).

In order to identify TFs that regulate SOCE-mediated gene expression in *Drosophila* neurons, we examined the upstream regions (up to 10 kb) of SOCE-regulated genes using motif enrichment analysis (*Zambelli et al., 2009*) for TF binding sites (*Figure 4A*). This analysis helped identify several putative TFs with enriched binding sites in the regulatory regions of SOCE-regulated genes (*Figure 4B*, *Figure 4—figure supplement 1*) which were then analysed for developmental expression trajectories (*Hu et al., 2017*) in the *Drosophila* CNS (*Figure 4B*, lower panel). Among the top 10 identified TFs, the homeobox TF Trl/GAF had the highest enrichment value, lowest p-value (*Figure 4B*), and was consistently expressed in the CNS through all developmental stages, with a distinct peak during pupal development (*Figure 4B*, lower panel). To test the functional significance of Trl/GAF, we tested flight in flies with *THD'*-specific knockdown of *Trl* using two independent RNAi lines (*Figure 4C*) as well as in existing *Trl* mutant combinations that were viable as adults (*Figure 4—figure supplement 2*). While homozygous *trl* null alleles are lethal, trans-heterozygotes of two different hypomorphic alleles were viable and exhibit significant flight deficits (*Figure 4C*, *Figure 4—figure supplement 2*). Moreover, the flight deficits caused by *Trl$^{RNAi}$* specifically in the *THD'* neurons could be rescued by *Set2* overexpression (*Figure 4C*, compare data in red and brown).

To further test Trl function, as an effector of intracellular Ca$^{2+}$ signalling and SOCE, we investigated genetic interactions of *trl$^{13C}$*, a hypomorphic mutant-recessive *Trl* allele, with an existing deficiency for the ER-Ca$^{2+}$ sensor STIM (*STIM$^{KO}$*), an essential activator of the Orai channel (*Wang et al., 2010*) as well as existing mutants for an intracellular ER-Ca$^{2+}$ release channel, the IP$_3$R (*Joshi et al.,*

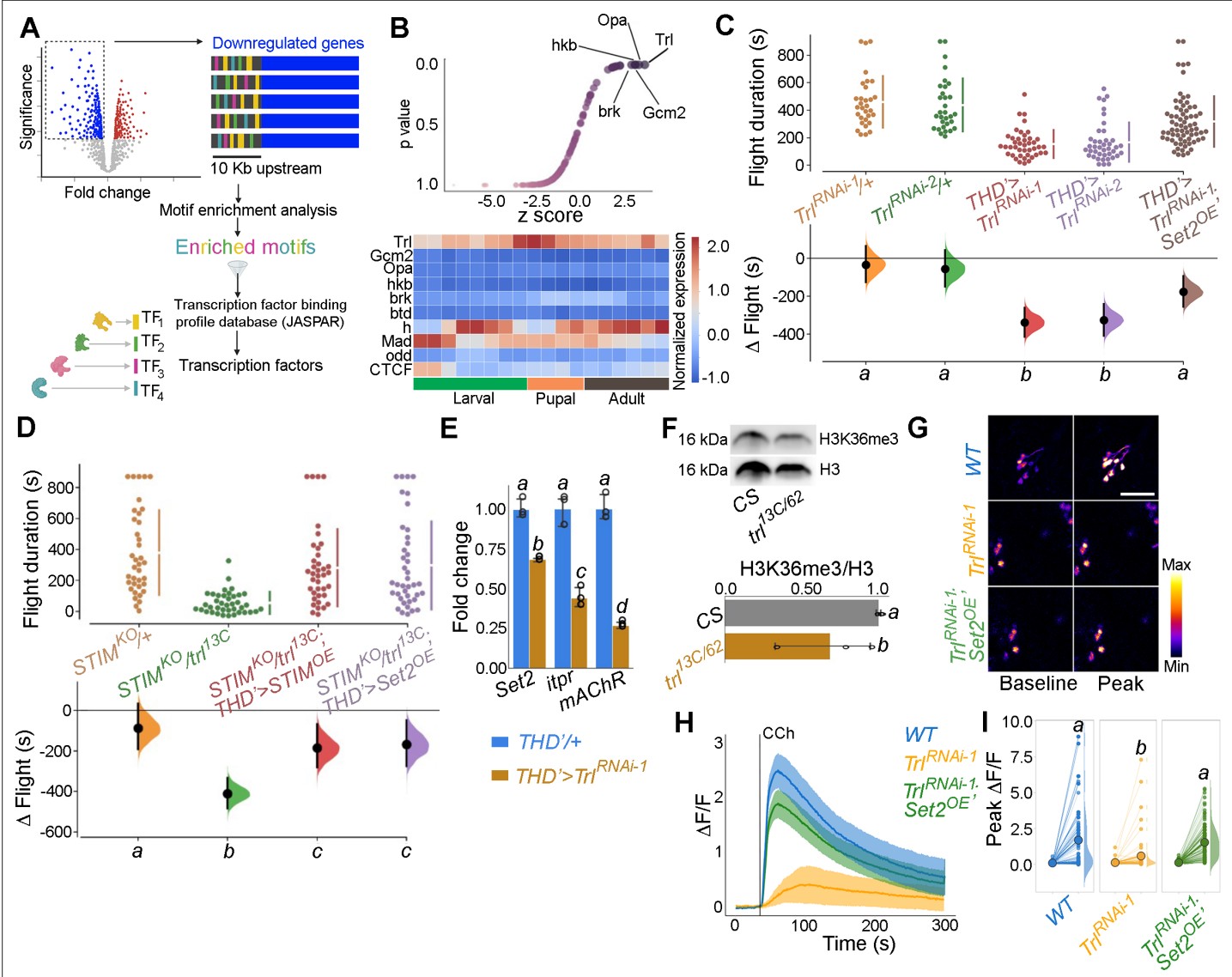

**Figure 4.** Identification of trithorax-like (Trl) as a store-operated Ca²⁺ entry (SOCE)-responsive transcription factor (TF). (**A**) Schematic of motif enrichment analysis for identification of putative SOCE-dependent TFs. (**B**) Candidate TFs identified (upper panel) and their expression through development (lower panel; modEncode; ***modENCODE Consortium et al., 2010***) suggests Trl as a top SOCE-responsive candidate TF. (**C**) Genetic depletion of *Trl* in *THD'* dopaminergic neurons (DANs) results in significant flight defects, which can be rescued by overexpression of *Set2*. (**D**) Heteroallelic combination of the *Trl* hypomorphic allele (*trl¹³C*) with a *STIM* deficiency causes significant flight deficits, which can be rescued by overexpression of *STIM* or *Set2*. (**E**) qRT-PCRs measured relative to *rp49* show reduced *Set2, itpr, and mAChR* levels in *THD'* DANs upon *Trl* knockdown with *Trl*ᴿᴺᴬⁱ⁻¹. Individual data points of four biological replicates are shown as circles and mean expression level as a bar plot (± SEM). The letters above indicate statistically indistinguishable groups after performing a Kruskal–Wallis test and a Mann–Whitney *U*-test (p<0.05). (**F**) A representative western blot (left) showing reduced H3K36me3 levels in a *Trl* mutant combination. Quantification of H3K36me3 from three biological replicates of *WT* (control) and *Trl* mutant brain lysates (panel on the right). Letters indicate statistically indistinguishable groups after performing a two-tailed *t*-test (p<0.05). (**G**) Knockdown of *Trl* in *THD'* neurons attenuates Ca²⁺ response to carbachol (CCh) as shown in representative images of GCaMP6m fluorescence quantified in (**H, I**). Scale bar = 10 µM. *Set2* overexpression in the background of *THD'>Trl*ᴿᴺᴬⁱ rescues the cholinergic response (**G–I**). Quantification of Ca²⁺ responses is from 10 or more brains per genotype and performed as described in the legend to ***Figure 3***. The letters above indicate statistically indistinguishable groups after performing a Kruskal–Wallis test and a Mann–Whitney *U*-test (p<0.05).

The online version of this article includes the following source data and figure supplement(s) for figure 4:

**Source data 1.** Raw data for TF motif enrichment analysis (B), flight assays (C, D), RT-qPCRs (E), western blot quantifications (F),and imaging quantifications (H, I).

**Figure supplement 1.** Motifs of the top ranked transcription factors (TFs) identified using motif enrichment.

*Figure 4 continued on next page*

*Figure 4 continued*

**Figure supplement 2.** Trithorax-like (Trl) is required for flight.

**Figure supplement 2—source data 1.** Raw data for flight assays.

**Figure supplement 3.** Genetic interaction between hypomorphic *Trl* alleles and *itpr* alleles results in significant flight defects.

**Figure supplement 3—source data 1.** Raw data for flight assays.

**Figure supplement 4.** Annotation of the *Set2* promoter revealed multiple trithorax-like (Trl)-binding sites.

*2004*). While $STIM^{KO}/+$ flies demonstrate normal flight bout durations, adding a single copy of the $trl^{13C}$ allele to the $STIM^{KO}$ heterozygotes ($STIM^{KO}/+$; $trl^{13C}/+$ trans-heterozygotes) resulted in significant flight deficits, which could be rescued by overexpression of either *STIM* or *Set2* (*Figure 4D*). The rescue of $STIM^{KO}/+$; $trl^{13C}/+$ with $STIM^{OE}$ (*Figure 4D*) indicates that SOCE, driven by $STIM^{OE}$, activates residual Trl encoded by a copy of $Trl^+$ in this genetic background ($trl^{13c}/+$) to rescue flight. The role of Trl as an SOCE-regulated TF is further supported by rescue of flight in $STIM^{KO}/+$; $trl13c/+$ flies by overexpression of *Set2* (*Figure 4D*). Set2 expression in *THD'* neurons was demonstrated earlier as requiring Orai-mediated $Ca^{2+}$ entry (*Figure 2B and C*). These genetic data are consistent with the positive feedback loop proposed earlier (*Figure 3F*). We hypothesise that Trl is non-functional upon expression of $Orai^{E180A}$ due to reduced SOCE. Loss of Trl function in turn downregulates *Set2* expression. A single copy of $trl^{13C}$ placed in combination with a single copy of various *itpr* mutant alleles also showed a significant reduction in the duration of flight bouts (*Figure 4—figure supplement 3*), further supporting a role for intracellular $Ca^{2+}$ signalling in Trl function. Flight deficits observed upon specific expression of $Trl^{RNAi}$ in *THD'* neurons (*Figure 4C*) indicate the direct requirement of Trl in fpDANs. Taken together, these findings provide good genetic evidence for Trl as an SOCE-responsive TF in *THD'* neurons. Due to the strong flight deficit observed by expression of $Orai^{E180A}$, we were unable to test genetic interactions of *Trl* with Orai directly. $Orai^{RNAi}$ strains exhibit off-target effects and *Orai* hypomorphs are unavailable.

The requirement of *Trl* for SOCE-dependent gene expression was tested directly by measuring *Set2* transcripts in FACS-sorted *THD'* neurons with knockdown of *Trl* ($THD'>Trl^{RNAi}$). *Set2* has 20 Trl binding sites in a 2 kb region upstream of the transcription start site (*Figure 4—figure supplement 4*) and knockdown of *Trl* resulted in downregulation of *Set2* (32%) and the Set2-regulated genes *itpr* (56%) and *mAchR* (73%; *Figure 4E*). Moreover, a trans-heterozygotic combination of hypomorphic Trl alleles ($trl^{13C/62}$) had markedly reduced levels of brain H3K36me3 (*Figure 4F*).

To test whether Trl drives cellular function in the fpDANs, we stimulated the mAChR with CCh and measured $Ca^{2+}$ responses in fpDANs with knockdown of *Trl*. Compared to *WT* fpDANs, which respond robustly to cholinergic stimulation, $Trl^{RNAi}$-expressing DANs exhibit strongly attenuated responses (*Figure 4G–I*). Overexpression of *Set2* in the $Trl^{RNAi}$ background rescued the cholinergic response (*Figure 4G–I*). The rescue of both flight (*Figure 4C and D*) and the cholinergic response (*Figure 4G–I*) by overexpression of *Set2* in flies with knockdown of *Trl* in *THD'* neurons ($THD'>Trl^{RNAi}$) confirms that Trl acts upstream of *Set2* to ensure optimal neuronal function and flight.

The results obtained so far indicate that Trl functions as an intermediary TF between SOCE and its downstream effector gene *Set2*. Overexpression of *Trl*, however, is unable to rescue the loss of flight caused by loss of SOCE ($THD'>Orai^{E180A}$; *Figure 5A*). Moreover, *Trl* transcript levels are unchanged in the $Orai^{E180A}$ condition (*Figure 5B*). These data could either mean that the phenotypes observed upon loss of Trl are independent of Orai-$Ca^{2+}$ entry or that Trl requires $Ca^{2+}$ influx through SOCE for its function to go from an 'inactive' form to an 'active' form (schematised in *Figure 5C*). We favour the latter interpretation because the key transcripts downregulated by expression of $Orai^{E180A}$ (*Set2*, *itpr*, and *mAchR*) were also downregulated upon knockdown of *Trl* in *THD'* neurons (see *Figures 3E and 4E*).

To understand how Orai-mediated $Ca^{2+}$ entry might activate Trl, we went back to previously known biochemical characterisation of Trl. Although Trl does not possess a defined $Ca^{2+}$ binding domain, it interacts with a diverse set of proteins, primarily through its BTB-POZ domain (*Figure 5—figure supplement 1*) as demonstrated earlier by affinity purification of Trl from embryonic extracts followed by high-throughput mass spectrometry (*Lomaev et al., 2017*). Among the identified interacting partners, we focused on kinases, keeping in mind an earlier study in *Drosophila* indicating phosphorylation of Trl at a threonine residue (T237; *Zhai et al., 2008*). Analysis of the Trl interactome revealed several kinases, which were expressed to varying extents in the *THD'* neurons (*Figure 5—figure supplement*

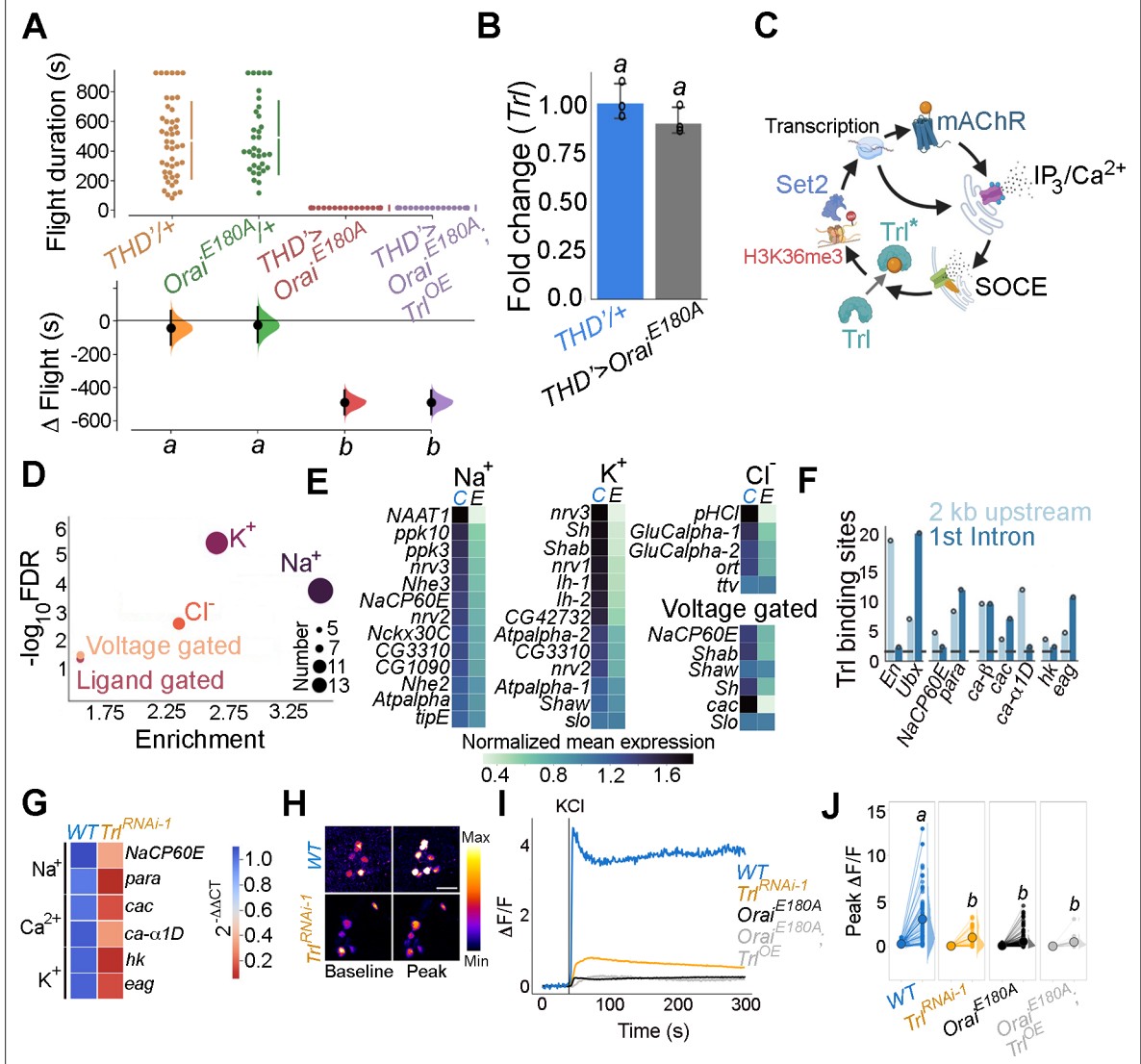

**Figure 5.** Trithorax-like (Trl) activity downstream of store-operated Ca²⁺ entry (SOCE) is required for *THD'* dopaminergic neuron (DAN) activity. (**A**) Overexpression of *WT Trl* is insufficient to rescue flight deficits in *THD'>OraiE^E180A* flies as evident from flight bout measurements in the indicated genotypes. Flight assay measurements are as described earlier. (**B**) *Trl* transcript levels are not altered in *THD'>OraiE^E180A* neurons with loss of SOCE. qRT-PCR data are measured relative to *rp49*. The bar plot indicates mean expression levels, with individual data points represented as circles. The letters above indicate statistically indistinguishable groups from three independent biological replicates after performing a two-tailed *t*-test (p<0.05). (**C**) Schematic with possible Ca²⁺- mediated activation of Trl downstream of Orai-mediated Ca²⁺ entry (SOCE). (**D**). Genes encoding ion channels are enriched among SOCE-responsive genes in *THD'* DANs as determined by Gene Ontology (GO) analysis. Circles of varying radii are scaled according to the number of genes enriched in that category. (**E**) Downregulation of individual ion channel genes depicted as a heatmap. (**F**) Number of Trl binding sites (GAGA repeats) shown as a bar plot in the regulatory regions (2 kb upstream or first intron) of key SOCE-responsive ion channel genes compared to known Trl targets (*En* and *Ubx*). The dashed line indicates the average expected number of Trl targets. (**G**) Heatmap of 2^-ΔΔCT values measured using qRT-PCRs from sorted *THD'* DANs with knockdown of *Trl*. (**H**) Representative images of KCl-induced depolarising responses in *THD'* DANs with knockdown of *Trl*. (scale bar = 10 μM) quantified in (**I, J**). Quantification of Ca²⁺ responses is from 10 or more brains per genotype and performed as described in the legend to *Figure 3*. The letters above indicate statistically indistinguishable groups after performing a Kruskal–Wallis test and a Mann–Whitney *U*-test (p<0.05).

The online version of this article includes the following source data and figure supplement(s) for figure 5:

**Source data 1.** Raw data for flight assays (A), RT-qPCRs (B), GO analysis (D), transcript quantfication from RNA-Seq (E, G), TF bindiing site analysis (F),and imaging quantifications (I, J).

**Figure supplement 1.** In silico analysis of trithorax-like (Trl) reveals a BTB/POZ domain, a Zn-finger domain, and a Q-rich domain.

*Figure 5 continued on next page*

*Figure 5 continued*

**Figure supplement 2.** Possible interacting partners of trithorax-like (Trl) enriched from an LC-MS dataset (*Lomaev et al., 2017*) and compared to their expression in *THD'* neurons as a scatter plot.

**Figure supplement 2—source data 1.** Raw data for Trl interacting proteins.

**Figure supplement 3.** *THD'*-specific inhibition of CamKII activity either by expression of an inhibitory Ala peptide or through RNAi downregulation results in significant flight deficits.

**Figure supplement 3—source data 1.** Raw data for flight assays.

**Figure supplement 4.** Expression of a dominant active Ca$^{2+}$-insensitive CamKII (*CamKII$^{T287D}$*) partially rescues flight in *THD'>OraiE$^{E180A}$* flies.

**Figure supplement 4—source data 1.** Raw data for flight assays.

2), including the Ca$^{2+}$-dependent CamKII, implicated earlier in flight in *THD'* neurons (*Ravi et al., 2018*). Loss of CamKII in *THD'* neurons by RNAi-mediated knockdown or through expression of a peptide inhibitor (Ala; *Mehren and Griffith, 2004*) resulted in significant flight deficits (*Figure 5—figure supplement 3*). To test whether CamKII activation downstream of SOCE is required for flight, we expressed a constitutively active version of CamKII (*T287D*; *Kadas et al., 2012*) in the background of *THD'>OraiE$^{E180A}$*. The phosphomimetic *CamKII$^{T287D}$* point mutation renders CamKII activity Ca$^{2+}$ independent (*Malik and Hodge, 2014*). Acute expression of *CamKII$^{T287D}$* in the *Orai$^{E180A}$* background using the TARGET system (*McGuire et al., 2004*) resulted in a weak rescue of flight when implemented from 72 to 96 hr APF (*Figure 5—figure supplement 4*), whereas overexpression of WT *CamKII* or a phosphorylation-incompetent allele (*CamKII$^{T287A}$*) failed to rescue flight (*Figure 5—figure supplement 4*). These data support a model (schematised in *Figure 5C*) wherein Ca$^{2+}$ influx through SOCE sustains Trl activation, in part through CamKII, leading to expression of *Set2* followed by increased levels of chromatin H3K36me3 marks that drive further downstream gene expression changes through a transcriptional feedback loop. Our data do not rule out activation of other Ca$^{2+}$-sensitive mechanisms for activation of Trl. Moreover, rescue of flight in Trl mutant/knockdown conditions by *STIM* overexpression (*Figure 4C and D*) suggests the presence of additional SOCE-responsive TFs in fpDANs.

## Trl function downstream of SOCE targets neuronal activity through changes in ion channel gene expression including VGCCs

Next we investigated the extent to which the identified SOCE-Trl-Set2 mechanism impacts fpDAN function. 'Ion transport' is among the top GO terms identified in SOCE-responsive genes (*Figure 2A*). Indeed, multiple classes of ion channel genes, including several Na$^+$, K$^+$, and Cl$^-$ channels, are downregulated in fpDANs upon expression of *Orai$^{E180A}$* (*Figure 5D and E*). Regulation of ion channel gene expression by Trl was indicated from the enrichment of Trl binding sites in regulatory regions of some ion channel loci (*Figure 5F*). We purified fpDANs with knockdown of *Trl* (*THD'>TrlRNAi$^{RNAi-1}$*) and measured expression of a few key voltage-gated ion channel genes (*Figure 5F*). *NaCP60E*, *para* (Na$^+$ channels), *cac*, *ca-α1D* (VGCC subunits), *hk*, and *eag* (outward-rectifying K$^+$ channels) are all downregulated between 0.6- and 0.9-fold upon *Trl* knockdown (*Figure 5G*) as well as upon loss of SOCE (*Figure 5B*). Taken together, these results indicate that the expression of key voltage-gated ion channel genes which are required for the depolarisation-mediated response, maintenance of electrical excitability, and neurotransmitter release is a focus of the transcriptional program set in place by SOCE-Trl-Set2.

To test the functional consequences of altered ion channel gene expression downstream of Trl, neuronal activity of the fpDANs was tested by means of KCl-evoked depolarisation. PPL1 DANs exhibited a robust Ca$^{2+}$ response that was lost upon knockdown of *Trl* (*Figure 5H–J*, data in orange). In consonance with the earlier behavioural experiments (*Figure 5A*), overexpression of *Trl$^+$* in *THD+* DANs did not rescue the KCl response in PPL1 DANs with loss of SOCE (*Orai$^{E180A}$*; *Trl$^{OE}$*, *Figure 5I and J*), further reinforcing the hypothesis that Trl activity in fpDANs is dependent on Ca$^{2+}$ entry through Orai.

Response to KCl is also lost in PPL1 neurons by knockdown of *Set2* and restored to a significant extent by overexpression of *Set2* in *THD'* DANs lacking SOCE (*Orai$^{E180A}$:Set2$^{OE}$*; *Figure 6A–C*). We confirmed that the KCl response of PPL1 neurons requires VGCC function either by treatment with nimodopine (*Xu and Lipscombe, 2001*), an L-type VGCC inhibitor, or by knockdown of a conserved VGCC subunit *cac* (*cac$^{RNAi}$*). Both forms of perturbation abrogated the depolarisation response to KCl

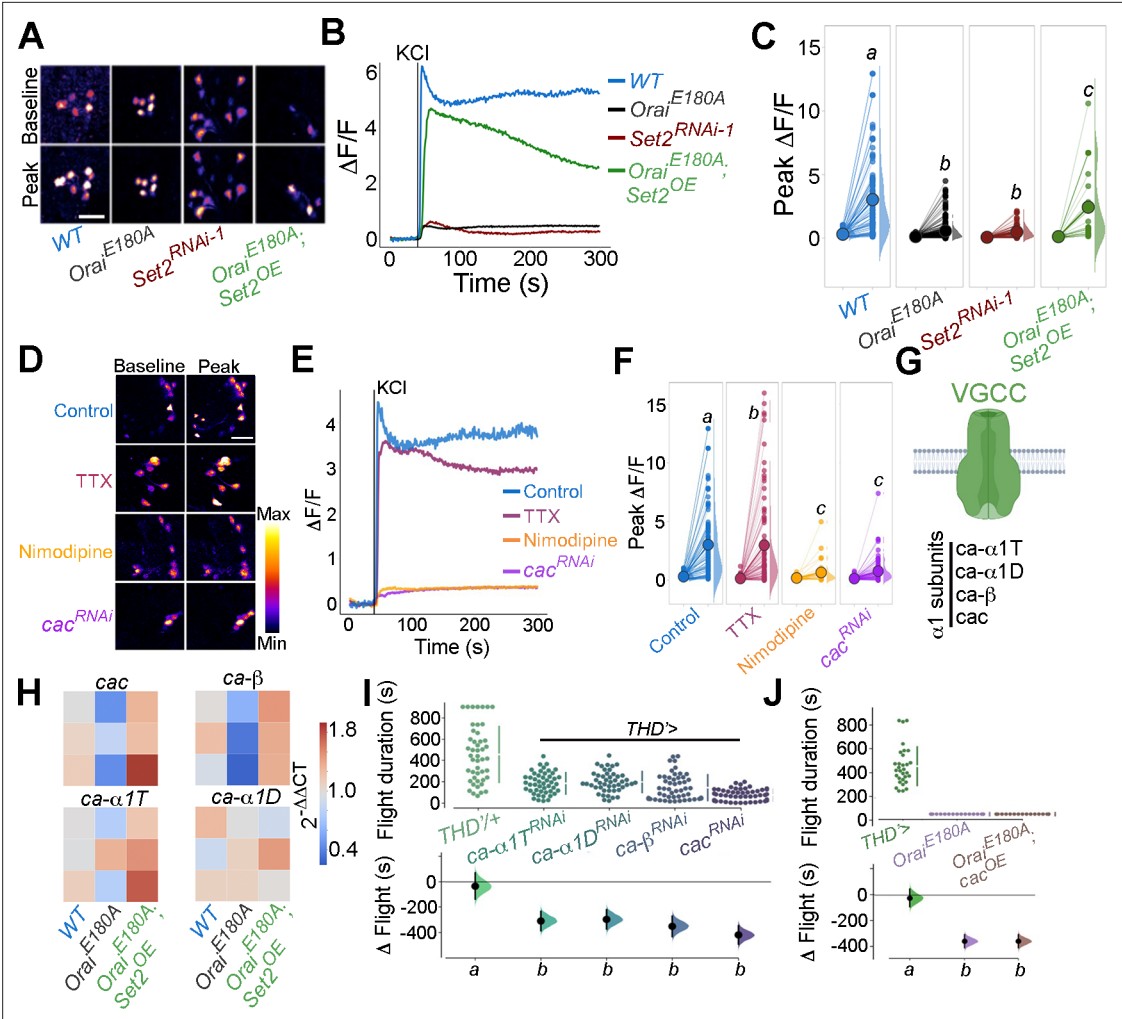

**Figure 6.** Flight-promoting central dopaminergic neuron (fpDAN) excitability requires Orai-mediated Ca$^{2+}$ entry acting through *Set2*-mediated VGCC gene expression. KCl-induced depolarising responses in *fp*DANs of the indicated genotypes measured using GCaMP6m indicate a requirement for Orai and *Set2*. Representative images (**A**). Scale bar = 10 μM, quantified in (**B**) and (**C**). KCl-evoked responses are modulated upon treatment with 10 μM tetrodotoxin (TTX) (magenta), 10 μM nimodipine (orange), and upon *cac$^{RNAi}$* (purple), representative images (scale bar = 10 μM) (**D**), quantified in (**E**). Median KCl-evoked GCaMP6m responses plotted as a function of time. The solid line indicates the time point of addition of 70 mM KCl. KCl responses are quantified as a paired plot of peak responses and (**F**) with letters representing statistically indistinguishable groups as measured using a Kruskal–Wallis test and post hoc Mann–Whitney *U*-test (p<0.05). Quantification of Ca$^{2+}$ responses is from 10 or more brains per genotype and performed as described in the legend to *Figure 3*. (**G**) Schematic of a typical VGCC and its constituent subunits. (**H**) Heatmap of 2^-ΔΔCT values measured using qRT-PCRs from sorted *THD'* DANs shows a reduction in the expression of VGCC subunit genes upon loss of Orai function and a rescue by *Set2* overexpression. (**I**) Flight assays representing the effect of various VGCC subunit gene RNAis showing flight defects to varying extents. Overexpression of the key VGCC subunit gene-*cac* is required (**I**) but not sufficient (**J**) for restoring flight defects caused by loss of *Orai* function. Flight assay measurements are as described earlier.

The online version of this article includes the following source data for figure 6:

**Source data 1.** Raw data for imaging quantitations (B, C, E, F), RT-qPCRs (H),and flight assays and (I, J).

in PPL1 neurons (**Figure 6D–F**, data in purple or orange, respectively). Moreover, the Ca$^{2+}$ entry upon KCl-mediated depolarisation in PPL1 neurons is a cell-autonomous property as evident by measuring the response after treatment with the Na$^+$ channel inhibitor, tetrodotoxin (TTX; **Figure 6D–F**, data in magenta).

Having confirmed that the Ca$^{2+}$ response towards KCl depolarisation requires VGCCs, we tested the expression level of key components of *Drosophila* VGCCs (**Figure 6G**) in *THD'* DANs, including *cacophony* (*cac*), *ca-α1D*, *ca-α1T*, and *ca-β*. All four subunits of *Drosophila* VGCC are downregulated upon loss of SOCE (*Orai$^{E180A}$*; **Figure 6H**) in *THD'* DANs, whereas overexpression of *Set2* in

the background of $Orai^{E180A}$ restored expression of the four VGCC subunits tested (*Figure 6H*), thus explaining recovery of the KCl response upon *Set2* overexpression in PPL1 neurons lacking SOCE ($Orai^{E180A}$; *Figure 6A–C*).

The functional significance of downregulation of VGCC subunits by the SOCE-Trl-Set2 pathway was tested directly by knockdown of the four VGCC subunits independently in PPL1 neurons followed by measurement of flight bout durations. Significant loss of flight was observed with knockdown of each subunit (*Figure 6I*) though in no case was the phenotype as strong as what is observed with loss of SOCE ($Orai^{E180A}$; *Figure 1D*). These data suggest that in addition to VGCC subunit expression, appropriate expression of other ion channels in the fpDANs is of functional significance. This idea is further supported by the observation that loss of flight upon loss of SOCE cannot be restored by over-expression of *cac* alone (*Figure 6J*), indicating that optimal neuronal activity requires an ensemble of genes, including ion channels, whose expression is regulated by SOCE-Trl-Set2.

### *THD'* DANs require SOCE for developmental maturation of neuronal activity

The functional relevance of changes in neuronal activity during pupal maturation of the PPL1-dependent flight circuit was tested next. Inhibition of *THD'* DANs from 72 to 96 hr APF using acute induction of the inward-rectifying K+ channel *Kir2.1* (*Johns et al., 1999*) or optogenetic inhibition through the hyperpolarising Cl- channel *GtACR2* (*Govorunova et al., 2015*) resulted in significant flight deficits (*Figure 7A*, data in dark or light green). Similar flight defects were recapitulated upon hyperactivation of *THD'* neurons through either optogenetic (*CsChrimson*; *Klapoetke et al., 2014*) or thermogenetic (*TrpA1*; *Viswanath et al., 2003*) stimulation (*Figure 7A*, data in orange or red), indicating that these neurons require balanced neuronal activity during a critical window in late pupal development, failing which the flight circuit malfunctions. To test whether restoring excitability in fpDANs lacking SOCE (*THD'>OraiE^{E180A}*) is sufficient to restore flight, we induced hyperexcitability by overexpressing *NachBac* (*Nitabach et al., 2006*), a bacterial Na+ channel. *NachBac* expression rescued both flight (*THD'>OraiE^{E180A}*; *Figure 7B*) and excitability (*Figure 7C–E*). A partial rescue of flight was also obtained by optogenetic stimulation of neuronal activity through activation of *THD'>OraiE^{E180A}* DANs using *CsChrimson* either 72–79 hr APF or 0–2 d post eclosion (*Figure 7E*). In this case, flight rescues were accompanied by a corresponding rescue of *CsChrimson* activation-induced Ca2+ entry (*Figure 7F and G*). Moreover, inducing neuronal hyperactivation with indirect methods such as excess K+ supplementation (*Figure 7—figure supplement 1*) or impeding glial K+ uptake, by genetic depletion of the glial K+ channel *sandman* (*Weiss et al., 2019*; *Figure 7—figure supplement 2*), partially rescued flight in animals lacking SOCE in the *THD'* DANs. Together, these results indicate that SOCE, through Trl and Set2 activity, determines activity in fpDAN during circuit maturation by regulating expression of voltage-gated ion channel genes, like *cac* (see 'Discussion').

fpDANs also require a balance between H3K36me3/H3K27me3-mediated epigenetic regulation (*Figure 2K*). To understand whether this epigenetic balance affects flight through modulation of neuronal excitability, we measured KCl-induced depolarising responses of PPL1 DANs in Orai-deficient animals (*THD'>OraiE^{E180A}*) fed on the H3K27me3 antagonist GSK343 (0.5 mM). GSK343-fed animals were sorted into two groups (fliers/non-fliers) on the basis of flight bout durations of greater or lesser than 30 s (*Figure 7—figure supplement 3*) and tested for responses to KCl (*Figure 7—figure supplement 3C*). *THD'>OraiE^{E180A}* flies fed on GSK343 showed a rescue in KCl responses, with a clearly enhanced response in the fliers compared to the non-fliers (*Figure 7—figure supplement 3C*), thereby reiterating the hypothesis that SOCE-mediated regulation of activity in these neurons acts through epigenetic regulation between opposing histone modifications.

### *THD'* DANs require SOCE-mediated gene expression for axonal arborisation and neuromodulatory dopamine release in the MB γ lobe

Next, we revisited how loss of activity during the critical maturation window of 72–96 hr APF and 0–2 d post-eclosion affects axonal arborisation of fpDANs that innervate the γ lobe. From previous work, we know that major axonal branches of the fpDANs reach the γ lobe normally in *THD'>OraiE^{E180A}* animals (*Figure 8A and B*; *Pathak et al., 2015*). Neuronal activity can drive changes in neurite complexity and axonal arborisation (*Depetris-Chauvin et al., 2011*) especially during critical developmental periods (*Sachse et al., 2007*). To understand whether Orai-mediated Ca2+ entry and

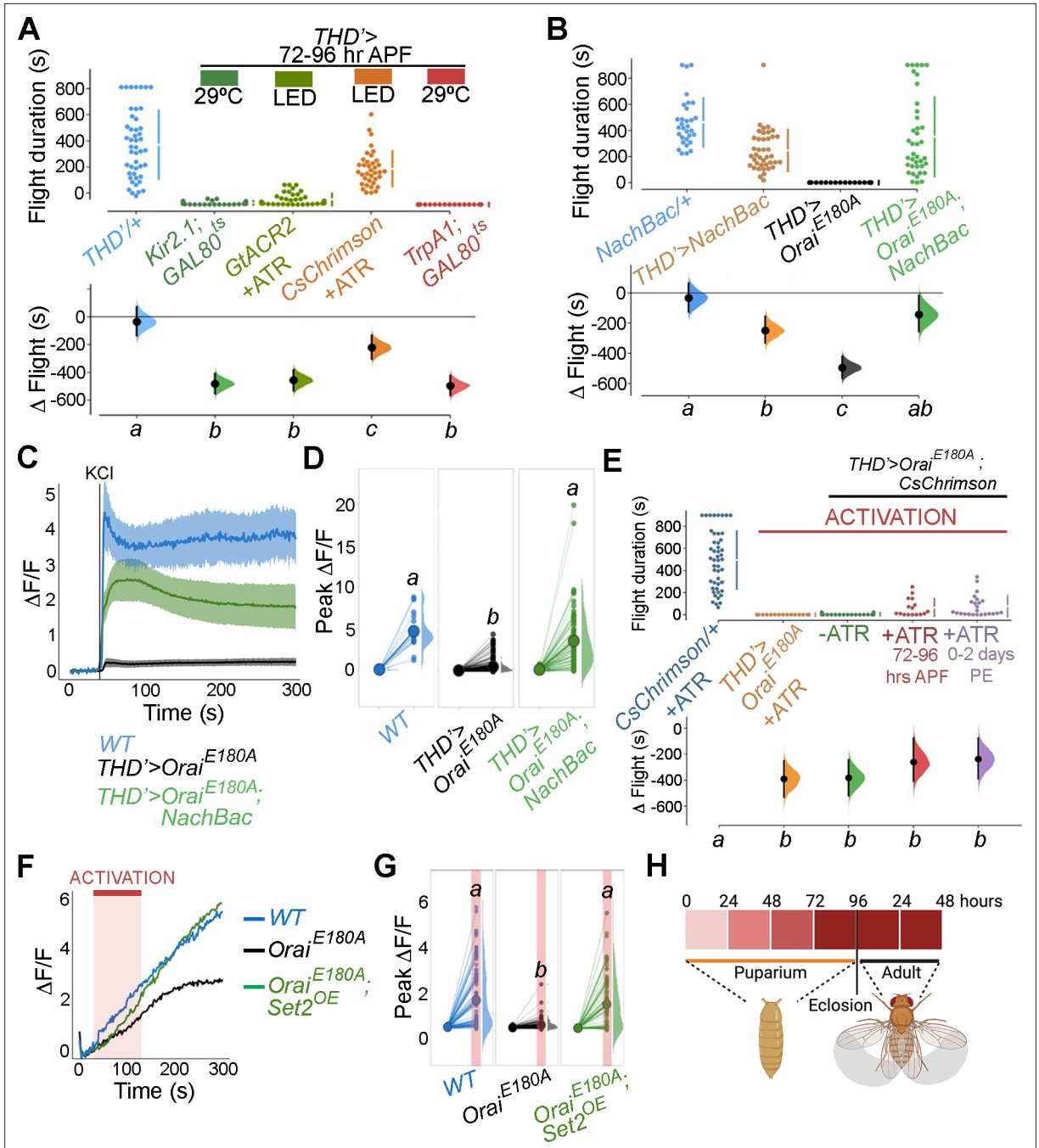

**Figure 7.** Store-operated Ca$^{2+}$ entry (SOCE)-mediated gene expression sets the excitability threshold during pupal development. (**A**) Altering excitability in *THD'* dopaminergic neurons (DANs) using *Kir2.1* or *GtACR2*-mediated inhibition or *CsChrimson* or *TrpA1* during the critical 72–96 hr after puparium formation (APF) developmental window results in significant flight defects. Orai loss-of-function phenotypes can be rescued by overexpression of *NachBac* or *CsChrimson* in terms of flight bout durations (**B, E**) and depolarising KCl responses (**C, D, F, G**). (**H**) Schematic of Orai-mediated Ca$^{2+}$ entry regulating expression of key genes that regulate the excitability threshold of dopaminergic neurons regulating flight during a critical developmental window. Flight assays are represented as described earlier (n > 30). Ca$^{2+}$ responses were quantified as described in the legend to *Figure 3* and were from 10 or more brains per genotype. Letters above each genotype represent statistically indistinguishable groups as measured using a Kruskal–Wallis test and post hoc Tukey test (p<0.05).

The online version of this article includes the following source data and figure supplement(s) for figure 7:

**Source data 1.** Raw data for flight assays (A, B, E), and imaging quantifications (C, D, F, G).

**Figure supplement 1.** Dietary KCl supplementation shows a minor rescue of flight in a dose-dependent manner in Orai-deficient (*THD'>OraiE180A*) animals.

*Figure 7 continued on next page*

*Figure 7 continued*

**Figure supplement 1—source data 1.** Raw data for flight assays.

**Figure supplement 2.** Inhibition of glial K[+] uptake by knockdown of the K[+] channel-*sandman* in the background of Orai deficiency in the dopaminergic neurons (DANs) shows a partial rescue in flight.

**Figure supplement 2—source data 1.** Raw data for flight assays.

**Figure supplement 3.** Loss of flight (**A**) and loss of Ca[2+] entry upon KCl depolarisation (**B**) in *THD'>Orai^E180A* flies are partially rescued upon feeding an H3K27me3 antagonist- GSK343 (0.5 mm).

**Figure supplement 3—source data 1.** Raw data for flight assays (A) and imaging quantifications (B, C).

downstream gene expression through Set2 affects this activity-driven parameter, we investigated the complexity of fpDAN presynaptic terminals within the γ2α'1 lobe MB using super-resolution micros-copy (*Figure 8A and B*). Striking changes in the neurite volume upon expression of *Orai^E180A* were observed. Importantly, these changes could be rescued to a significant extent by restoring either *Set2* (*Orai^E180A; Set2^OE*) or by inducing hyperactivity through *NachBac* expression (*Orai^E180A; NachBac^OE*; *Figure 8C and D*).

To understand whether reduced axonal arborisation within the γ lobe has functional consequences, we measured CCh-evoked DA release at the γ2α'1 region of the MB (*Figure 8A*, red arrow). For this purpose, we used the dopamine sensor *GRAB-DA* (*Sun et al., 2018*). Loss of SOCE (*THD'>OraiE^E180A*) attenuated DA release (*Figure 8E–G*), which could be partially rescued by overexpression of *Set2*. Taken together these data identify an SOCE-dependent gene regulation mechanism acting through the TF Trl, and the histone modifier Set2 for timely expression of genes that impact diverse aspects of neuronal function, including neuronal activity, axonal arborisation, and sustained neurotransmitter release (schematised in *Figure 8H*).

## Discussion

Over the course of development, neurons define an excitability set point within a dynamic range (*Truszkowski and Aizenman, 2015*), which stabilises existing connections (*Mayseless et al., 2023*) and enables circuit maturation (*Johnson-Venkatesh et al., 2015*). The setting of this threshold for individual classes of neurons is based on the relative expression of various ion channels. In this study, we identify an essential role for the store-operated Ca[2+] channel *Orai* in determining ion channel expression and neuronal activity in a subset of DANs central to an MB circuit for *Drosophila* flight. Orai-mediated Ca[2+] entry, initiated by neuromodulatory acetylcholine signals, is required for amplifi-cation of a signalling cascade, which begins by activation of the homeo-box TF Trl, followed by upreg-ulation of several genes, including *Set2*, a gene that encodes an enzyme for an activating epigenetic mark, H3K36me3. Set2 in turn establishes a transcriptional feedback loop to drive expression of key neuronal signalling genes, including a repertoire of voltage-gated ion channel genes required for neuronal activity in mature PPL1 DANs (schematised in *Figure 9*).

### SOCE supports gene expression transitions during critical developmental windows

Studies in the *Drosophila* MB have identified spontaneous bouts of voltage-gated Ca[2+] channel-mediated neuronal activity through as yet unknown mechanisms, which occur during early adult-hood driving refinement and maturation of behaviours such as associative learning (*Leinwand and Scott, 2021*). Our studies here, on a subset of 21–23 central DANs (*Figure 1B and C*) of which some send projections to the MB and regulate flight behaviour, provide a mechanistic explanation for neuromodulation-dependent intracellular signalling culminating in transcriptional maturation of a circuit supporting adult flight. In the critical developmental window of 72–96 hr APF, a cohort of SOCE-responsive genes, which include a range of ion channels, undergo a wave of induction. Loss of SOCE at this stage thus renders these neurons functionally incompetent (*Figures 3B–D, 6G–I and 7D–F*) and abrogates flight (*Figure 1D*). Developmentally assembled neuronal circuits require experi-ence/activity-dependent maturation (*Akin and Zipursky, 2020*). The requirement of SOCE for flight circuit function during the early post-eclosion phase (0–2 d) suggests that maturation of the flight circuit extends into early adulthood, where presumably feedback from circuit activity may further

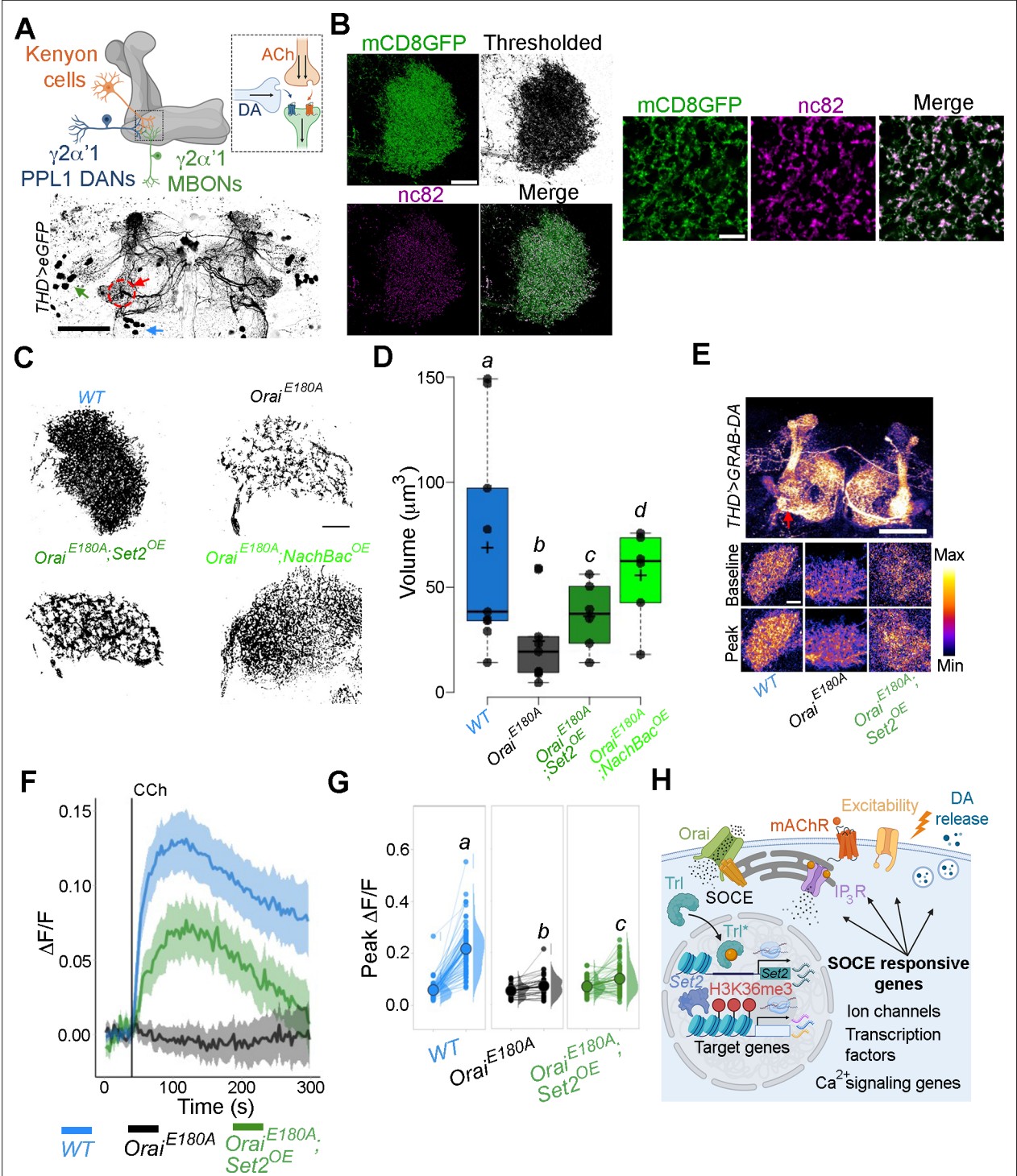

**Figure 8.** *THD'* dopaminergic neurons (DANs) require store-operated Ca²⁺ entry (SOCE)-mediated gene expression for axonal arborisation and neuromodulatory dopamine release in the mushroom body (MB) γ lobe. (**A**) Schematic of a KC-DAN-MBON tripartite synapse at the γ2α'1 MB lobe (upper), innervated by flight-promoting central dopaminergic neurons (fpDANs) (lower). The γ2α'1 MB lobe is marked by a red circle and arrow, and the fpDAN clusters are marked with a green arrow (PPL1) and blue arrow (PPM3). Scale bar = 50 μM. (**B**) Representative images of the MB γ lobe marked with THD'-driven mCD8GFP (green) and immunolabelled with anti-nc82 (Brp) antibody (magenta) to mark the presynaptic terminals. Scale bar = 5 μm. (**C**) Representative optical sections through the MB γ lobe from the indicated genotypes. Changes in axonal arborisation observed were quantified as total projection volume in (**D, E**). Carbachol (CCh)-evoked DA release measured at the γ2α'1 MB lobe using the GRAB-DA sensor expressed in *THD'* DANs. Representative GRAB-DA images of *THD'* DANs are shown with baseline and peak-evoked responses in the indicated genotypes. Scale bar =

*Figure 8 continued on next page*

*Figure 8 continued*

10 μM. (**F**) Median GRAB-DA responses plotted as a function of time. A shaded region around the solid line represents the 95% confidence interval from 10 or more brains per genotype. (**G**) Individual cellular responses depicted as a paired plot of peak responses where different letters above indicate statistically distinguishable groups after performing a Kruskal–Wallis test and a Mann–Whitney *U*-test. (**H**) Schematic describing how Orai-mediated $Ca^{2+}$ entry regulates expression of key genes that control neuronal activity in fpDANs.

The online version of this article includes the following source data for figure 8:

**Source data 1.** Raw data for imaging quantifications (D, F, G).

refine synaptic strengths (**Sugie et al., 2018**). Expression of *Orai^{E180A}* in 5-day-old adults had no effect on flight (**Figure 1E**), indicating that either SOCE is not required for maintaining gene expression in adults or that wildtype Orai channel proteins from late pupal and early adults perdure for long periods extending into adulthood. Loss of SOCE in adult brains beyond 5–7 d, when Orai protein turnover might be expected, has not been assessed so far. The absence of any visible changes to primary

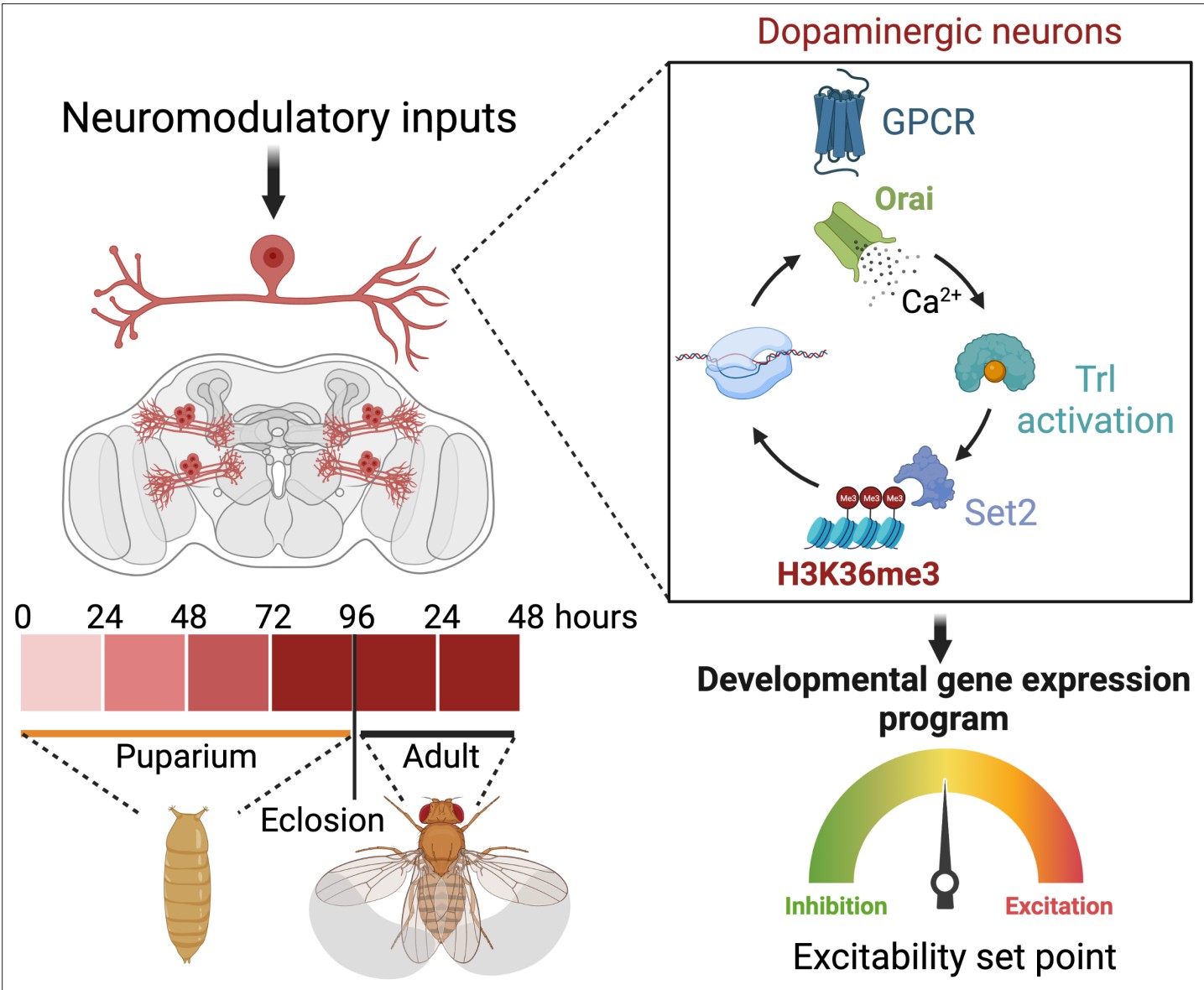

**Figure 9.** Schematic outline of the role of store-operated $Ca^{2+}$ entry (SOCE) in determining excitability and function of dopaminergic neurons required for flight circuit development. SOCE acts through a transcriptional feedback loop including the transcription factor trithorax-like (Trl), and the H3K36 methyltransferase Set2 to induce the expression of genes required for neuronal excitability and flight.

neurite patterning suggests that SOCE in the MB-DANs works primarily to refine synaptic function at presynaptic terminals within the MB lobe. Taken together, our findings identify a role for SOCE in restructuring the neuronal transcriptome during a critical developmental transition, which facilitates key changes in neuronal function required for circuit formation and adult behaviour (schematised in *Figure 9*). In summary, our findings here identify Orai-driven Ca$^{2+}$ entry as a key signalling step for driving coordinated gene expression enabling the maturation of nascent neuronal circuits and sustenance of adult behaviour.

## SOCE regulates a balance between competing epigenetic signatures

Chromatin structure and function are dynamic over the course of brain development (*Kishi and Gotoh, 2018*). One way neurons achieve dynamic spatio-temporal control over developmental gene expression is via epigenetic mechanisms such as post-translational histone modifications (*Geng et al., 2021*). Cells possess an extensive toolkit of histone modifiers, with characterised effects on transcriptional output by either activating or repressing gene expression (*Bannister and Kouzarides, 2011*). However, relatively little is known about how modifiers of histone marks are in turn regulated to bring about the requisite changes in neuronal gene expression over developmental timescales.

The SET-domain containing family of histone modifiers (*Figure 2B*), including H3K36me3 methyltransferase Set2 (*Figure 2C*), identified here appears to function as key transcriptional effectors-induced downstream of neuromodulatory inputs that stimulate SOCE through Orai during pupal and adult maturation of fpDANs. SOCE-driven *Set2*-mediated H3K36me3 enhances the expression of key GPCRS, components of intracellular Ca$^{2+}$ signalling (*Figure 2A*) and ion channels (*Figure 6A and B*) for optimal dopaminergic neuron function (*Figure 7D–F*) and flight (*Figure 1C*). Rescue of flight by genetic depletion of an epigenetic 'eraser' *Kdm4B*, an H3K36me3 demethylase (*Figure 2D*), suggests that a balance between perdurance of H3K36me3 and its removal is actively maintained on the expressed genes. Interestingly, another member of the SET domain family, *E(z)*, is upregulated upon loss of SOCE (*Figure 2B*). *E(z)* is a component of the *Drosophila* PRC2 complex, which represses gene expression through H2K27me3 (*Margueron and Reinberg, 2011*). Structural studies on *Drosophila* nucleosomes have elucidated that the H3K36me3 modification allosterically inhibits the PRC2 complex activity (*Finogenova et al., 2020*). Our findings support a model wherein at key developmental stages SOCE determines the level of these competing epigenetic signatures, thus allowing gene expression by enhancing Set2-mediated H3K36me3 (*Figure 2H*). Loss of SOCE leads to deficient Set2-mediated H3K36me3 (*Figure 2E*), a likely shifting of the balance in favour of PRC2-mediated H3K27me3 and overall repression in gene expression (as observed in *Figure 1—figure supplements 8 and 9*). Feeding SOCE-deficient animals a pharmacological inhibitor of PRC2 (GSK434) led to a significant rescue in behaviour (*Figure 2J*) presumably by disinhibition of H3K27me3-mediated gene repression. Though flight initiation is rescued in a dose-dependent manner, the duration of flight bouts is not sustained (*Figure 2—figure supplement 8*), indicating that suppression of H3K27me3 marks in the SOCE-deficient flies is partial. These findings indicate additional roles for other histone modifications such as Set1-mediated H3K4me3, which may work in concert to regulate developmental gene expression programs stimulated by SOCE. Future studies directed at a comprehensive cataloguing of multiple epigenetic signatures in defined neuronal subsets using cell-specific techniques should help elucidate these mechanisms.

## Trl as an SOCE-responsive transcription factor

SOCE-mediated regulation of gene expression has been reported in mammalian neural progenitor cells (*Somasundaram et al., 2014*) and *Drosophila* pupal neurons (*Pathak et al., 2015*; *Richhariya et al., 2017*), through as yet unknown mechanisms. A combination of motif enrichment analysis over regulatory regions of SOCE-responsive genes and expression enrichment analysis in cells of interest helped generate a list of putative SOCE-responsive TFs (*Figure 4A and B*). Among these, we experimentally validated Trl/trithorax-like/GAGA factor (GAF) as a TF required for maturation of excitability in fpDANs using genetic tools (*Figure 4*). We found that loss of *Trl* in the fpDANs results in significant flight deficits, which could be partially rescued through excess STIM (which raises SOCE). This indicates that increased Ca$^{2+}$ entry through Orai either activates residual Trl or alternates SOCE-responsive TFs to induce *Set2* expression and rescue downstream gene expression essential for flight.

Although Trl function has been reported in the context of early embryonic development, where it has been implicated in zygotic genome activation (*Gaskill et al., 2021*) (ZGA), expression of *Hox* genes (*Shimojima et al., 2003*), dosage compensation (*Greenberg et al., 2004*), and expression of a *Drosophila* voltage-gated calcium channel subunit in germ cell development (*Dorogova et al., 2014*), this is the first ever report of its role in regulating neuronal gene expression in the context of SOCE and circuit maturation. The molecular mechanism by which Orai-mediated Ca$^{2+}$ entry leads to activation of Trl needs further elucidation. Our finding that CaMKII hyperactivation partially rescues flight deficits caused by loss of Orai function, taken together with the bioinformatic prediction for CaMKII-mediated Trl phosphorylation, needs to be experimentally verified. Future studies could be directed towards looking at the interactions between these two proteins using in vitro biochemical assays.

In vitro studies on *Drosophila* Trl reveal that it utilises a Q-rich intrinsically disordered domain (IDD) to self-multimerise or interact with a wide range of accessory proteins (*Wilkins and Lis, 1999*). Multimeric complexes of Trl reside on chromatin with an exceptionally long residence time and maintain chromatin in an 'open' state (*Tang et al., 2022*). Trl has also been reported to directly interact with components of the transcription initiation and elongation complex (*Chopra et al., 2008*; *Li et al., 2013*), support long-distance promoter–enhancer interactions (*Mahmoudi et al., 2003*), mediate active ATP-dependent chromatin remodelling by maintaining nucleosome-free stretches (*Tsukiyama et al., 1994*), and regulate global gene expression by controlling transcriptional stalling and pausing (*Tsai et al., 2016*). A recent study demonstrates a role for Trl in chromatin folding in *Drosophila* neurons to enable cell type-specific gene expression (*Mohana et al., 2023*). SOCE-dependent gene regulation may rely upon multiple TFs acting in concert in a cell type and developmental stage-specific contexts, of which Trl may be just one. Future studies directed at other possible TFs downstream of SOCE would be of interest.

## SOCE and the control of neuronal activity and cholinergic neuromodulation of a tripartite synapse in the MB

Neurons undergo maturation of their electrical properties with a gradual increase in depolarising responses and synaptic transmission over the course of pupal development (*Järvilehto and Finell, 1983*). Here, we show that a subset of DANs require a balance of ion channels for optimal excitation and inhibition essential for adult function. Neuromodulatory signals, such as acetylcholine, are essential for acquiring this balance and act via an SOCE-Trl-Set2 mechanism during late pupal and early adult stages. The ability of *Set2* overexpression to rescue SOCE-deficient phenotypes (*Figure 6A–C*) indicates that anomalous gene expression is the underlying basis of this phenotype. Indeed, SOCE-deficient DANs show reduced expression of an ensemble of voltage-gated ion channel genes (*Figure 5E and F*), including genes encoding the outward-rectifying K$^+$ channel *shaw*, slow-inactivating voltage-gated K$^+$ channel *shaker*, and a Ca$^{2+}$-gated K$^+$ channel *slowpoke* and *cacophony*, a subunit of the voltage-gated Ca$^{2+}$ channel (*Figure 6G*). Ion channels like *Shaker* and *slowpoke* play an important role in establishing resting membrane potential (*Singh and Wu, 1990*), neuronal repolarisation (*Lichtinghagen et al., 1990*), and mediating afterhyperpolarisation (AHP) after repeated bouts of activity (*Ping et al., 2011*). Though DANs require *Set2*-mediated expression of *cacophony* for excitability, *cacophony* is in itself insufficient to rescue the loss in excitability and flight deficits upon loss of SOCE (*Figure 6J*). Future experiments that directly visualise electrophysiological responses of fpDANs from different genotypes would help resolve the role of Cac and other ion channels and determine their individual contributions to THD' neuronal activity and excitability. Both flight and KCl depolarisation in SOCE-deficient neurons could, however, be rescued by expression of a heterologous depolarisation-activated Na$^+$ channel NachBac (*Nitabach et al., 2006*; *Figure 7B–D*), which increases neuronal excitability by stimulating a low-threshold positive feedback loop and to a lesser extent by CsChrimson-mediated optogenetic induction of neuronal activity (*Figure 7E*). The poorer rescue by optogenetic activation may stem from the fact that sustained bursts of activity are non-physiological and can deplete the readily releasable pool of neurotransmitters (*Arrigoni and Saper, 2014*; *Kaeser and Regehr, 2017*). Other indirect forms of inducing neuronal hyperactivation (*Figure 7—figure supplement 1*) in SOCE-deficient neurons also achieved minor rescues in flight. These results identify SOCE as a key driver downstream of developmentally salient neuromodulatory signals for expression of an ion channels suite that enables generation of intrinsic electrical properties and functional maturation of the PPL1 DANs and the flight circuit.

## Limitations of this study

Direct measurements of SOCE are generally performed in cultured cells by depletion of ER-store $Ca^{2+}$ in '0' $Ca^{2+}$ medium, followed by $Ca^{2+}$-add back to measure SOCE. In this study, we were unable to perform similar SOCE measurements from the fpDANs because they consist of 16–19 neurons in each hemisphere (PPL1 are 10–12 and PPM3 are 6–7 cells; *Pathak et al., 2015*) and identifying these few neurons was not technically feasible in culture. Measuring SOCE from these neurons in vivo was not possible due to the presence of abundant extracellular $Ca^{2+}$ in the brain. Due to these reasons, we have relied upon using CCh to elicit $IP_3$-mediated $Ca^{2+}$ release and SOCE as a proxy for in vivo SOCE. In previous studies, we have shown that CCh treatment of cultured *Drosophila* neurons elicits $IP_3$-mediated $Ca^{2+}$ release and SOCE (*Agrawal et al., 2010*). Moreover, expression of $Orai^{E180A}$ completely blocks SOCE as measured in primary cultures of DANs (*Pathak et al., 2015*) and CCh-induced $IP_3$-mediated $Ca^{2+}$ release is tightly coupled to SOCE in *Drosophila* neurons (*Venkiteswaran and Hasan, 2009*; *Chakraborty et al., 2016*; *Chakraborty and Hasan, 2017*). We posit that our measurements of CCh-evoked changes in cellular $Ca^{2+}$ reflect a composite of $IP_3$-mediated $Ca^{2+}$ release and SOCE.

Although we have provided compelling genetic evidence for Trl as an SOCE-responsive TF, the detailed biochemical basis of Trl activation was beyond the scope of this study. We propose phosphorylation by CamKII as a possible mechanism that needs further investigation.

## Materials and methods

### Fly maintenance

*Drosophila* strains were grown on standard cornmeal medium consisting of 80 g corn flour, 20 g glucose, 40 g sugar, 15 g yeast extract, 4 ml propionic acid, 5 ml *p*-hydroxybenzoic acid methyl ester in ethanol, and 5 ml ortho-butyric acid in a total volume of 1 l (ND) at 25°C under a light–dark cycle of 12 hr and 12 hr. Canton S was used as wildtype throughout. Mixed sex populations were used for all experiments. Several fly stocks used in this study were sourced using FlyBase (https://flybase.org), which is supported by a grant from the National Human Genome Research Institute at the US National Institutes of Health (#U41 HG000739), the British Medical Research Council (#MR/N030117/1), and FlyBase users from across the world. The stocks were obtained from the Bloomington Drosophila Stock Centre (BDSC) supported by NIH P40OD018537. All fly stocks used and their sources are listed in *Supplementary file 1b*.

### Flight assays

Flight assays were performed as previously described (*Manjila and Hasan, 2018*). Briefly, flies aged 3–5 d of either sex were tested in batches of 8–10 flies, and a minimum of 30 flies were tested for each genotype. Adult flies were anaesthetised on ice for 2–3 min and then tethered between at the head–thorax junction using a thin metal wire and nail polish. Post recovery at room temperature for 2–3 min, an air puff was provided as stimulus to initiate flight. Flight duration was recorded for each fly for 15 min. For all control genotypes, GAL4 or UAS strains were crossed to the wildtype strain, *Canton S*. Flight assays are represented in the form of a swarm plot, wherein each dot represents a flight bout duration for a single fly. The colours indicate different genotypes. The Δ Flight parameter refers to the mean difference for comparisons against the shared CS control which is shown as a Cumming estimation plot (*Ho et al., 2018*). On the lower axes, mean differences are plotted as bootstrap sampling distributions. Each mean difference is depicted as a dot. Each 95% confidence interval is indicated by the ends of the vertical error bars. Letters beneath each distribution refer to statistically indistinguishable groups after performing a Kruskal–Wallis test followed by a post hoc Mann–Whitney *U*-test (p<0.05).

### FACS

Fluorescence-activated cell sorting (FACS) was used to enrich eGFP-labelled DANs from pupal/adult. The following genotypes were used for sorting: wildtype (*THD'GAL4>UAS-eGFP*), $Orai^{E180A}$ (*THD'GAL4>OraiE^{E180A}*), Set2^{IR-1}(*THD'GAL4>Set2^{IR-1}*), $Orai^{E180A}$; Set2^{OE} (*THD'GAL4>OraiE^{E180A}; Set2^{OE}*), Set2^{OE} (*THD'GAL4>Set2^{OE}*), and Trl^{IR-1} (*THD'GAL4>TrlIR^{IR-1}*). Approximately 100 pupae per sample were washed in 1× PBS and 70% ethanol. Pupal/adult CNSs were dissected in Schneider's medium (Thermo Fisher Scientific) supplemented with 10% fetal bovine serum, 2% PenStrep, 0.02 mM insulin,

20 mM glutamine, and 0.04 mg/ml glutathione. Post dissection, the CNSs were treated with an enzyme solution (0.75 g/l collagenase and 0.4 g/l dispase in Rinaldini's solution [8 mg/ml NaCl, 0.2 mg/ml KCl, 0.05 mg/ml $NaH_2PO_4$, 1 mg/ml $NaHCO_3$, and 0.1 mg/ml glucose]) at room temperature for 30 min. They were then washed and resuspended in ice-cold Schneider's medium and gently triturated several times using a pipette tip to obtain a single-cell suspension. This suspension was then passed through a 40 mm mesh filter to remove clumps and kept on ice until sorting (less than 1 hr). Flow cytometry was performed on a FACS Aria Fusion cell sorter (BD Biosciences) with a 100 mm nozzle at 60 psi. The threshold for GFP-positive cells was set using dissociated neurons from a non-GFP-expressing wildtype strain (*Canton S*). The same gating parameters were used to sort other genotypes in the experiment. GFP-positive cells were collected directly in TRIzol and then frozen immediately in dry ice until further processing. Details for all reagents used are listed in *Supplementary file 1b*.

## RNA-seq

RNA from at least 600 sorted *THD'* DANs from the relevant genotypes, expressing *UAS-eGFP*, was subjected to 14 cycles of PCR amplification (SMARTer Seq V4 Ultra Low Input RNA Kit; Takara Bio). Then, 1 ng of amplified RNA was used to prepare cDNA libraries (Nextera XT DNA library preparation kit; Illumina). cDNA libraries for four biological replicates for both control (*THD'GAL4/+*) and experimental (*THD'GAL4>UAS-Orai^{E180A}*) genotypes were run on a Hiseq2500 platform. Then, 50–70 million unpaired sequencing reads per sample were aligned to the dm6 release of the *Drosophila* genome using HISAT2 (*Kim et al., 2015*; *Kim et al., 2019*) and an overall alignment rate of 95.2–96.8% was obtained for all samples. Featurecounts (*Liao et al., 2014*) was used to assign the mapped sequence reads to the genome and obtain read counts. Differential expression analysis was performed using three independent methods: DESeq2 (*Love et al., 2014*), limma-voom (*Ritchie et al., 2015*), and edgeR (*Robinson et al., 2010*). A fold change cut-off of a minimum twofold change was used. Significance cut-off was set at an FDR-corrected p-value of 0.05 for DESeq2 and edgeR. Volcanoplots were generated using VolcaNoseR (https://huygens.science.uva.nl/). Comparison of gene lists and generation of Venn diagrams was performed using Whitehead BaRC public tools (http://jura.wi.mit.edu/bioc/tools/).GO analysis for molecular function was performed using DAVID (*Huang et al., 2009*). Developmental gene expression levels were measured for downregulated genes using FlyBase (*Larkin et al., 2021*) and DGET (*Hu et al., 2017*; https://www.flyrnai.org/tools/dget/web/) and were plotted as a heatmap using ClustVis (*Metsalu and Vilo, 2015*; https://biit.cs.ut.ee/clustvis/).

## qRT-PCRs

CNSs of the appropriate genotype and age were dissected in 1× phosphate-buffered saline (PBS; 137 mM NaCl, 2.7 mM KCl, 10 mM $Na_2HPO_4$, and 1.8 mM $KH_2PO_4$) prepared in double-distilled water treated with DEPC. Each sample consisted of five CNS homogenised in 500 µl of TRIzol (Ambion, Thermo Fisher Scientific) per sample. At least three biological replicate samples were made for each genotype. After homogenisation, the sample was kept on ice and either processed further within 30 min or stored at −80°C for up to 4 wk before processing. RNA was isolated following the manufacturer's protocol. Purity of the isolated RNA was estimated by NanoDrop spectrophotometer (Thermo Fisher Scientific) and integrity was checked by running it on a 1% Tris-EDTA agarose gel. Approximately 100 ng of total RNA was used per sample for cDNA synthesis. DNAse treatment and first-strand synthesis were performed as described previously (*Pathak et al., 2015*). Quantitative real-time PCRs (qPCRs) were performed in a total volume of 10 µl with Kapa SYBR Fast qPCR kit (KAPA Biosystems) on an ABI 7500 fast machine operated with ABI 7500 software (Applied Biosystems). Technical duplicates were performed for each qPCR reaction. The fold change of gene expression in any experimental condition relative to wildtype was calculated as 2−ΔΔCt, where $\Delta\Delta C_t$ = [$C_t$ (target gene) −$C_t$ (rp49)] Expt. − [$C_t$ (target gene) − $C_t$ (rp49)]. Primers specific for rp49 and ac5c were used as internal controls. Sequences of all primers used are provided in Table S1c.

## Functional imaging

Adult brains were dissected in adult hemolymph-like saline (AHL: 108 mM NaCl, 5 mM KCl, 2 mM $CaCl_2$, 8.2 mM $MgCl_2$, 4 mM $NaHCO_3$, 1 mM $NaH_2PO_4$, 5 mM trehalose, 10 mM sucrose, 5 mM Tris, pH 7.5) after embedded in a drop of 0.1% low-melting agarose (Invitrogen). Embedded brains were bathed in AHL and then subjected to functional imaging. The genetically encoded calcium sensor

GCaMP6m (*Chen et al., 2013*) was used to record changes in intracellular cytosolic $Ca^{2+}$ in response to stimulation with CCh or KCl. The GRAB-DA sensor (*Sun et al., 2018*) was used to measure evoked dopamine release. Images were taken as a time series on an xy plane at an interval of 1 s using a ×20 objective with an NA of 0.7 on an Olympus FV3000 inverted confocal microscope. A 488 nm laser line was used to record GCaMP6m/GRAB-DA measurements. All live-imaging experiments were performed with at least 10 independent brain preparations. For measuring evoked responses, KCl/ CCh was added on top of the samples. For optogenetics experiments, flies were reared on medium supplemented with 200 µM all trans retinal (ATR), following which neuronal activation was achieved using a 633 nm laser line to stimulate *CsChrimson* (*Klapoetke et al., 2014*) while simultaneously acquiring images with the 488 nm laser line, and the images were acquired every 1 s. The raw images were extracted using FIJI (*Schindelin et al., 2012*; based on ImageJ version 2.1.0/1.53c). ΔF/F was calculated from selected regions of interest (ROIs) using the formula ΔF/F = (Ft – F0)/F0, where Ft is the fluorescence at time t and F0 is baseline fluorescence corresponding to the average fluorescence over the first 40 time frames. Mean ΔF/F time lapses were plotted using PlotTwist (*Goedhart, 2020*; https://huygens.science.uva.nl/PlotTwist/). A shaded error bar around the mean indicates the 95% confidence interval for CCh (50 µM) or KCl (70 mM) responses. Peaks for individual cellular responses for each genotype were calculated from before or after the point of stimulation using Microsoft Excel and plotted as a paired plot using SuperPlotsOfData (*Goedhart and Pollard, 2021*; https://huygens. science.uva.nl/SuperPlotsOfData/). Boxplots were plotted using PlotsOfData (*Postma and Goedhart, 2019*; https://huygens.science.uva.nl/PlotsOfData/).

## Immunohistochemistry

Immunostaining of larval *Drosophila* brains was performed as described previously (*Daniels et al., 2008*). Briefly, adult brains were dissected in 1× PBS and fixed with 4% paraformaldehyde. The dissected brains were washed 3–4 times with 0.2% phosphate buffer, pH 7.2 containing 0.2% Triton-X 100 (PTX), and blocked with 0.2% PTX containing 5% normal goat serum (NGS) for 2 hr at room temperature. Respective primary antibodies were incubated overnight (14–16 hr) at 4°C. After washing 3–4 times with 0.2% PTX at room temperature, they were incubated in the respective secondary antibodies for 2 hr at room temperature. The following primary antibodies were used: chick anti-GFP antibody (1:10,000; A6455, Life Technologies), mouse anti-nc82 (anti-brp) antibody (1:50), rabbit anti-H3K36me3, and mouse anti-RFP. Secondary antibodies were used at a dilution of 1:400 as follows: anti-rabbit Alexa Fluor 488 (#A11008, Life Technologies) and anti-mouse Alexa Fluor 488 (#A11001, Life Technologies). Confocal images were obtained on the Olympus Confocal FV1000 microscope (Olympus) with a ×40, 1.3 NA objective or with a ×60, 1.4 NA objective. Imaging for axonal arbors within the MB γ lobe was performed on the Zeiss LSM980 system with AiryScan 2. Images were visualised using either the FV10-ASW 4.0 viewer (Olympus) or FIJI (*Schindelin et al., 2012*). Details for all reagents used are listed in Table S1b.

## Western blotting

Adult CNSs of appropriate genotypes were dissected in ice-cold PBS. Between 5 and 10 brains were homogenised in 50 µl of NETN buffer (100 mM NaCl, 20 mM Tris–Cl [pH 8.0], 0.5 mM EDTA, 0.5% Triton-X-100, 1× Protease inhibitor cocktail [Roche]). The homogenate (10–15 µl) was run on an 8% SDS-polyacrylamide gel. The protein was transferred to a PVDF membrane by standard semi-dry transfer protocols (10 V for 10 min). The membrane was incubated in the primary antibody overnight at 4°C. Primary antibodies were used at the following dilutions: rabbit anti-H3K36me3 (1:5000, Abcam, ab9050) and rabbit anti-H3 (1:5000, Abcam, ab12079). Secondary antibodies conjugated with horseradish peroxidase were used at dilution of 1:3000 (anti-rabbit HRP; 32260, Thermo Scientific). Protein was detected by a chemiluminescent reaction (WesternBright ECL, Advansta K12045-20). Blots were first probed for H3K36me3, stripped with a solution of 3% glacial acetic acid for 10 min, followed by re-probing with the anti-H3 antibody.

## In silico ChIP-seq analysis

H3K36me3 enrichment data was obtained from a ChIP-chip dataset (ID_301) generated in *Drosophila* ML-DmBG3-c2 cells submitted to modEncode (*Celniker et al., 2009*; *modENCODE Consortium et al., 2010*). Enrichment scores for genomic regions were calculated using 'computematrix' and

plotted as a tag density plot using 'plotHeatmap' from deeptools2 (*Ramírez et al., 2016*). All genes were scaled to 2 kb with a flanking region of 250 bp on either end. A 50 bp length of non-overlapping bins was used for averaging the score over each region length. Genes were sorted based on mean enrichment scores and displayed on the heatmap in descending order.

### Statistical tests

Non-parametric tests were employed to test significance for data that did not follow a normal distribution. Significant differences between experimental genotypes and relevant controls were tested either with the Kruskal–Wallis test followed by Dunn's multiple-comparison test (for multiple comparisons) or with Mann–Whitney *U*-tests (for pairwise comparisons). Data with normal distribution were tested by Student's *t*-test. Statistical tests and p-values are mentioned in each figure legend. For calculation and representation of effect size, estimationstats.com was used.

## Acknowledgements

We acknowledge the Central Imaging and Flow Cytometry Facility (CIFF, NCBS) for maintenance of microscopes and the *Drosophila* facility (Flyfacility, NCBS) for Fly stock maintenance and development of transgenics. We thank Nandashree KS for helping with super-resolution imaging on the Zeiss 980 Airyscan system. This work was funded by grant no. BT/PR28450/MED/122/166/2018 from the Department of Biotechnology, Govt. of India, and core support by NCBS, TIFR. RM and SR received graduate student fellowships from NCBS, TIFR.

## Additional information

### Funding

| Funder | Grant reference number | Author |
| --- | --- | --- |
| Department of Biotechnology, Ministry of Science and Technology, India | BT/PR28450/MED/122/2018 | Gaiti Hasan |

The funders had no role in study design, data collection and interpretation, or the decision to submit the work for publication.

### Author contributions

Rishav Mitra, Data curation, Software, Formal analysis, Validation, Investigation, Visualization, Methodology, Writing – original draft, Writing – review and editing; Shlesha Richhariya, Investigation; Gaiti Hasan, Conceptualization, Supervision, Funding acquisition, Writing – original draft, Project administration, Writing – review and editing

### Author ORCIDs

Rishav Mitra http://orcid.org/0000-0001-6840-6160
Gaiti Hasan http://orcid.org/0000-0001-7194-383X

### Ethics

This study involved invertebrates (Drosophila melanogaster) and was performed in strict accordance with institutional biosafety and animal handling guidelines.

Joint Public Review https://doi.org/10.7554/eLife.88808.4.sa1
Author Response https://doi.org/10.7554/eLife.88808.4.sa2

## Additional files

### Supplementary files

• Supplementary file 1. List of fly stocks (a), reagents (b), and primer sequences (c) used in this

article.

• MDAR checklist

## Data availability

Sequencing data have been deposited in GEO under accession codes GSE2301342. Source data for each figure and figure supplements have been provided. All reagent information has been listed. A list of all primer sequences used and *Drosophila* stocks has been provided.

The following dataset was generated:

| Author(s) | Year | Dataset title | Dataset URL | Database and Identifier |
|---|---|---|---|---|
| Mitra R, Richhariya S, Hasan G | 2024 | Orai-mediated calcium entry regulates the excitability of central dopaminergic neurons | https://www.ncbi.nlm.nih.gov/geo/query/acc.cgi?&acc=GSE230134 | NCBI Gene Expression Omnibus, GSE230134 |

The following previously published datasets were used:

| Author(s) | Year | Dataset title | Dataset URL | Database and Identifier |
|---|---|---|---|---|
| Karpen G, Elgin S, Gortchakov A, Shanower G, Tolstorukov M, Kharchenko P, Kuroda M, Pirrotta V, Park P, Minoda A, Riddle N, Schwartz Y, Alekseyenko A, Kennedy C | 2013 | H3K36me3 abcam.OR Head Nuclei.Solexa | https://www.ncbi.nlm.nih.gov/geo/query/acc.cgi?acc=GSE47280 | NCBI Gene Expression Omnibus, GSE47280 |
| Karpen G, Elgin S, Gortchakov A, Shanower G, Tolstorukov M, Kharchenko P, Kuroda M, Pirrotta V, Park P, Minoda A, Riddle N, Schwartz Y, Alekseyenko A, Kennedy C | 2013 | H3K27me3 (Abcam lot3). OR Head Nuclei.Solexa | https://www.ncbi.nlm.nih.gov/geo/query/acc.cgi?acc=GSE47319 | NCBI Gene Expression Omnibus, GSE47319 |
| Graveley BR, Brooks AN, Carlson JW | 2011 | BDGP modENCODE *D. melanogaster* Transcriptome Sequencing | https://www.ncbi.nlm.nih.gov/bioproject/PRJNA75285 | NCBI BioProject, PRJNA75285 |

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
