## [Editor Report · eLife assessment]

In *Drosophila melanogaster*, the SOCE channel Orai is required for the development of flight-promoting dopaminergic neurons. The Hasan laboratory has previously shown that disabling Orai function impairs *Drosophila* flight due to aberrant neuronal development at the pupal stage. In this **fundamental** study, Mitra et al. show that SOCE drives a transcriptional feedback loop via the homeobox transcription factor, 'Trithorax-like' (Trl), and histone modifiers, Set2 and E(z), to regulate the expression of key genes required for the function of dopaminergic flight neurons, including the muscarinic acetylcholine receptor and the inositol 1,4,5-trisphosphate receptor. This **solid** study is carefully performed with validated methodology and most of the analyses are rigorous.

---

## [Referee Report · Joint Public Review]

In this study, Mitra and coworkers extend their previous analyses of the functional role of Orai in the excitability of central dopaminergic neurons in *Drosophila*. The authors show that a dominant-negative mutant of Orai (OraiE180A) significantly alters the gene expression profile of flight-promoting dopaminergic neurons (fpDANs), including that of Set2, E(z), and Trl, thereby shifting the level of epigenetic signatures that modulate gene expression. The Orai-Trl-Set2 pathway modulates the expression of voltage gated calcium channels, which, in turn, are involved in dopamine release. The study is generally well-done, is in-depth, and comprehensive. The finding that SOCE regulates a wide range of neuronal genes necessary for neuronal excitability and effector signaling by controlling chromatin remodelling genes is a noteworthy discovery.

The authors have adequately answered the previous concerns.

---

## [Author Response]

The following is the authors’ response to the previous reviews.

We thank the reviewers for collectively highlighting our study as “interesting and timely” and as making significant advances regarding the functional role of Orai in the activity of central dopaminergic neurons underlying the development of *Drosophila* flight behaviour. We hope that based on the revisions detailed below the data supporting our findings will be considered complete.

**Reviewer 1:**
In this revision, the authors have addressed most points using text changes but there is still one important issue that continues to be inadequately addressed. This relates to point 1.If Set2 is acting downstream of SOCE, it is not clear to me how STIM1 over expression rescues Set2-dependent downstream responses in flies that do not have Set2. It seems that if STIM1 over-expression, which would presumably enhance SOCE, largely rescues Set2-dependent effector responses in the Set2RNAi flies, then the proposed pathway cannot be true (because if Set2 is downstream of SOCE, it shouldn't matter whether SOCE is boosted in flies that lack Set2). This discrepancy is not explained. Does STIM1 over-expression somehow restore Set2 expression in the Set2RNAi flies?

Ans: Based on the requirement of Orai-mediated Ca2+ entry for Set2 expression (THD’>OraiE180A neurons, Figure 2C) we had indeed proposed that rescue of flight in Set2RNAi flies by STIMOE is because Set2 expression in Set2RNAi flies is restored by STIMOE. However, we agree that this has not been tested experimentally. Since these data are supportive but not essential to our findings here, we have removed data demonstrating flight rescue of Set2RNAi by STIMOE from Figure 2 – supplement 5 and associated text from the revised manuscript. We plan to investigate the effect of STIMOE on Set2 in the context of *Drosophila* dopaminergic neurons in the future.

**Reviewer 2:**
The manuscript analyses the functional role of Orai in the excitability of central dopaminergic neurons in *Drosophila*. The authors answer the previous concerns, but several important issues have not been experimentally tested. Especially, the lack of characterization of SOCE or calcium release from the intracellular calcium stores limits considerably the impact of the study. They comment on a number of technical problems but, taking into account the nature of the study, based on Orai and SOCE, the lack of these experimental data reduces the relevance of the study. Below are some specific comments:1. The response to question 1 is unconvincing. The authors do not demonstrate experimentally that STIM over-expression enhances SOCE or how excess SOCE might overcome the loss of SET2.

Ans: The reason we have not performed experiments in this manuscript to investigate SOCE in STIM overexpression condition is two-fold. Firstly, extensive characterisation of SOCE by STIM overexpression in *Drosophila* pupal neurons forms part of an earlier publication (Chakraborty and Hasan, Front. Mol. Neurosci, 2017). A graph from Chakraborty and Hasan, 2017 where SOCE was measured in primary cultures of pupal neurons from an IP3R mutant (S224F/G1891S) of Drosophila. Reduced SOCE in IP3R mutant neurons (red trace) was restored by overexpression of STIM (black trace). The green trace is of wild-type neurons with STIM overexpression and the grey trace with STIMRNAi. Similar experiments were performed with Orai+STIM overexpression and the rescue in SOCE was compared with STIM overexpression in pupal neurons of wild type and IP3R mutant S224F/G1891S. See Chakraborty and Hasan, 2017 (Front. Mol. Neurosci. 10:111. doi: 10.3389/fnmol.2017.00111)

1. Secondly, rescue by STIMOE is supportive but not essential to the findings of this manuscript which relate primarily to the analysis of an Orai-dependent transcriptional feed-back mechanism acting via Trl and Set2 in flight promoting dopaminergic neurons (See Fig 2C where we demonstrate that OraiE180A expression in THD’ neurons brings down Set2 expression).

We agree that we have not demonstrated how loss of Set2 can be compensated by STIM overexpression. Therefore, we have now removed the supplementary data relating to STIM rescue of Set2RNAi (THD’>Set2RNAi; STIMOE) flight phenotypes since as mentioned above it was supportive but not essential to the main theme of the manuscript. Consistent with this, we have also removed rescue of flight in TrlRNAi by STIMOE (Figure 4C).

2. The authors do not present a characterization of SOCE in the cells investigated expressing native Orai or the dominant negative OraiE180A mutant yet. They comment on some technical problems for in situ determination or using culture cells but, apparently, in previous studies they have reported some results.

Ans: We respectfully submit that characterisation of SOCE in cells expressing native Orai and OraiE180A from primary cultures of *Drosophila* pupal dopaminergic neurons, form part of an earlier publication (Pathak, T., et al., (2015). The Journal of Neuroscience, 35, 13784–13799. https://doi.org/10.1523/jneurosci.1680-15.2015). As mentioned in lines 80-84 the dopaminergic neurons studied here (THD’) are a subset of the dopaminergic neurons studied in the Pathak et al., 2015 publication (TH). As evident in Figure 2 panels B-D expression of OraiE180A in dopaminergic neurons abrogates SOCE.

In this study we have focused on identifying the molecular mechanism by which OraiE180A expression and concomitant loss of cellular Ca2+ signals (Figure 3B, 3C) affects dopaminergic neuron function. In lines 270-274 (page 10) we have stated the technical reason why Ca2+ measurements made in this study from ex-vivo brain preps measure a composite of ER-Ca2+ release and SOCE. Our observation that the measured Ca2+ response is significantly attenuated in cells expressing OraiE180A leads us to the conclusion that we are indeed measuring an SOCE component in the ex-vivo brain preps. This is also explained in ‘Limitations of the study’.

3. Concerning the question about the STIM:Orai stoichiometry the authors answer that "We agree that STIM-Orai stoichiometry is essential for SOCE, and propose that the rescue backgrounds possess sufficient WT Orai, which is recruited by the excess STIM to mediate the rescue"; however, again, this is not experimentally tested.

Ans: To address this point we have now measured relative stoichiometries of STIM and Orai mRNA by qPCR under WT conditions in *Drosophila* THD’ neurons at 72 hr APF. The observed stoichiometry as per these measurements is STIM:Orai = 1.6:1 (~8:5). These data are in relative agreement with the normalised read counts of STIM and Orai in THD’ neurons in the RNAseq performed and described in Fig 1F. The qPCR (A) and RNAseq (B) measures of STIM and Orai are appended below.

**Author response image 1. sa2fig1:** 

In comparison to the numerous studies investigating structural, biophysical and cellular characterisation of Orai channels in heterologous systems, there are fewer studies which have traced systemic implications of Orai function through multiple tiers of investigation including organismal behaviour. Leveraging the wealth of genetic resources available in *Drosophila*, we have attempted this here. While we respectfully agree that questions pertaining to the stoichiometries of STIM/Orai proteins are indeed relevant to cellular regulation of SOCE, we submit they may be better suited for investigation in heterologous systems involving cell culture, or with in-vitro systems with purified recombinant proteins, or indeed using computational and modelling approaches. None of these methods fall within the scope of our current investigation which is to understand how by Orai-mediated Ca2+ entry regulates developmental maturation of *Drosophila* flight promoting dopaminergic neurons.